# Paradoxical mTORC1-Dependent microRNA-mediated Translation Repression in the Nucleus Accumbens of Male Mice Consuming Alcohol Attenuates Glycolysis

Yann Ehinger [1], Sophie Laguesse[1,3], Khanhky Phamluong[1], Alexandra Salvi [1], Yoshitaka J. Sei[2], Zachary W. Hoisington[1], Drishti Soneja [1], Sowmya Gunasekaran[1], Ken Nakamura [1,2] & Dorit Ron [1] ✉

mTORC1 promotes protein translation, learning and memory, and neuroadaptations that underlie alcohol use disorder (AUD). The mechanisms underlying alcohol-mediated mTORC1-dependent neuroadaptations that drive AUD are not well understood. We report that activation of mTORC1 in the nucleus accumbens (NAc) D1 neurons of male mice consuming alcohol results in paradoxical mTORC1-dependent repression of translation of transcripts, including Aldolase A, an essential enzyme in glycolysis. We further show that mTORC1-dependent Aldolase A translation repression in D1 neurons is mediated through upregulation of miR-34a-5p expression. Alcohol-mediated mTORC1 repression of Aldolase A translation in D1 neurons inhibits glycolysis in the NAc. Finally, we report that overexpression of miR-34a-5p in D1 NAc neurons increases, whereas systemic administration of L-lactate, the final product of glycolysis, attenuates excessive alcohol intake. Our data suggest that alcohol promotes paradoxical actions of mTORC1 on translation and glycolysis which in turn drive excessive alcohol use.

The mechanistic target of rapamycin (mTOR) is a ubiquitously expressed serine and threonine kinase that, when localized with specific accessory proteins, including the adaptor protein raptor, is termed mTORC1 (mTOR complex 1)[1]. mTORC1 is activated by metabolism-relevant stimuli such as amino acids, lipid, oxygen, nutrients such as glucose as well as through the activation of growth factor signaling[1,2]. mTORC1 plays a central role in cell growth by promoting the translation of a subset of mRNAs to proteins, as well as well as by enhancing lipid and nucleotide signaling[1–3]. In contrast, mTORC1 also contributes to catabolism through the suppression of lysosomal biogenesis autophagy[2] and to aging[1]. mTORC1 also plays a crucial role in sensing and regulating feeding and fasting processes[1,3]. In the central nervous system, mTORC1 is known for its actions to promote the translation of dendritic spine proteins[4,5]. Specifically, upon activation mTORC1, phosphorylates its substrates the p70 ribosomal S6 kinase (S6K) and the eukaryotic translation initiation factor 4E binding protein (4E-BP)[6]. S6K then phosphorylates its substrate S6[6], and these phosphorylation events promote the assembly of the translation initiation complex to initiate cap-dependent and independent mRNA translation of transcripts[6]. mTORC1 together with its substrates are localized to ribosomes and the whole translation machinery is found in both cell body and dendrites[6]. Not surprisingly, mTORC1 plays an important role in synaptic plasticity, learning and memory[5,7,8].

Because of its important role in cellular homeostasis, dysregulation of mTORC1 functions results in pathologies such as cancer, obesity, liver and pancreatic abnormalities[1] as well as

[1]Department of Neurology, University of California, San Francisco, San Francisco, California, USA. [2]Gladstone Institute of Neurological Disease, Gladstone Institutes, San Francisco, California, USA. [3]Present address: S.L GIGA-Neurosciences, University of Liege, Liege, Belgium. ✉e-mail: dorit.ron@ucsf.edu

neurodegenerative, neurodevelopmental diseases and psychiatric disorders[5,9], including addiction[10].

Research on the mechanism(s) by which mTORC1 drives adverse phenotypes associated with drugs of abuse has not been explored much. We found that the signaling cascade upstream of mTORC1,

consisting of the small G protein H-Ras[11] and the kinases PI3K and AKT[12] is activated in the nucleus accumbens (NAc) of rodents consuming large quantities of alcohol. We further showed that pharmacological inhibition of H-Ras, PI3K, and AKT or knockdown of H-Ras in the NAc reduces excessive alcohol intake[11,12].

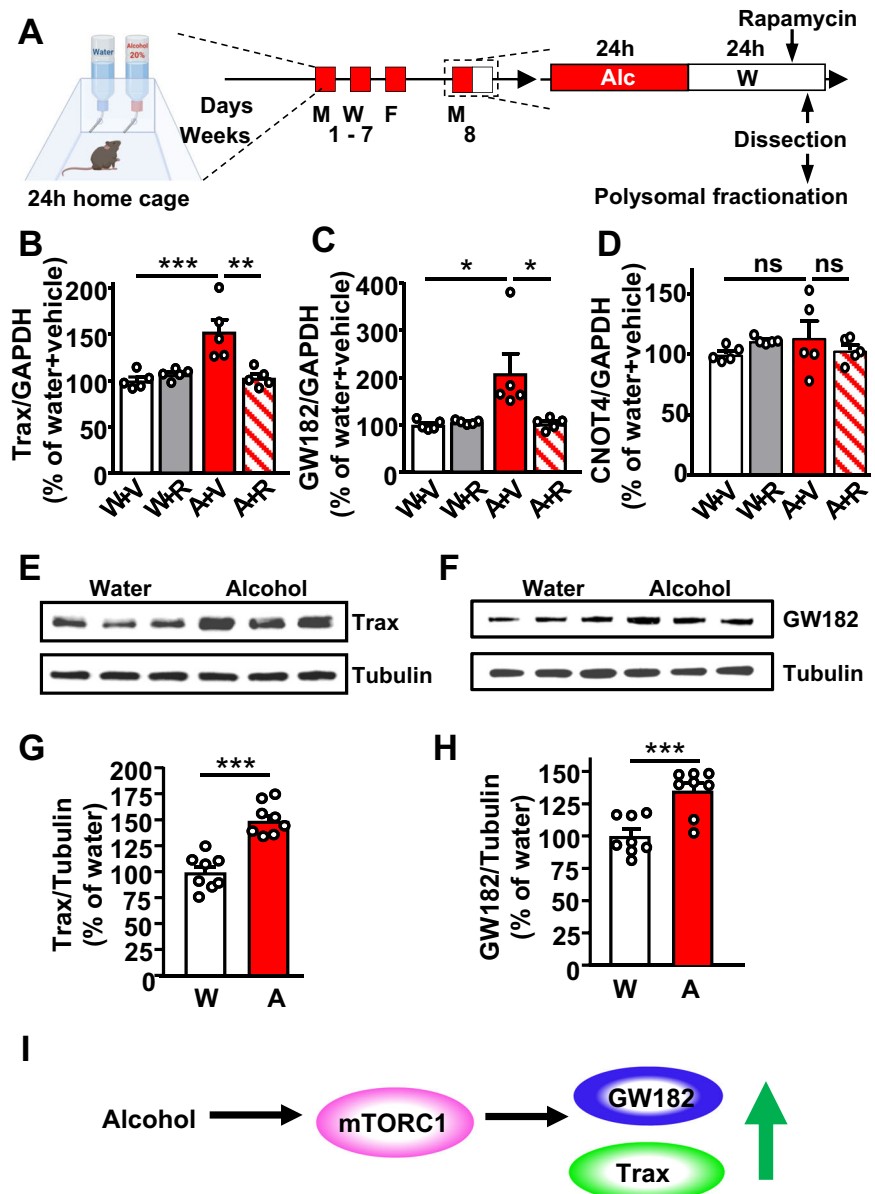

**Fig. 1 | Alcohol via mTORC1 increases the translation of Trax and GW182 but not CNOT4 in the NAc. A** Mice underwent 7 weeks of IA20%2BC (Supplementary Table 1). Control animals had access to 2 water bottles. Three hours before the end of the last alcohol withdrawal period, mice were systemically injected with 20 mg/kg rapamycin or vehicle. The NAc was dissected from each of the four mouse groups (water+vehicle in white, water+rapamycin in gray, alcohol+vehicle in red, alcohol+rapamycin in red and hatched) at the end of the last alcohol with-drawal period, were subjected to polysomal fractionation and RT-qPCR analysis. Created in BioRender (2025) https://BioRender.com/uvam6j0. **B–D** Polysomal RNA levels of Trax (**B**), GW182 (**C**) and CNOT4 (**D**) were measured by RT-qPCR. Each data point represents an average of 3 technical replicates. Data are presented as the average ratio of each transcript to GAPDH ± SEM and expressed as % of water +vehicle. *$p < 0.05$, **$p < 0.01$, ***$p < 0.001$, ns: non-significant. $n = 5$ mice per group. Significance was determined using Two-way ANOVA followed by Tukey's multiple comparisons test. (**B**) Alcohol x Rapamycin: $F_{(1, 16)} = 13.86$, $p = 0.0019$, effect of Alcohol: $F_{(1, 16)} = 10.23$, $p = 0.0056$, Effect of Rapamycin: $F_{(1, 16)} = 7.957$, $p = 0.0123$;

water and alcohol within the vehicle group, $p = 0.0008$, vehicle and rapamycin within the alcohol group, $p = 0.0014$. **C** Alcohol x Rapamycin: $F_{(1, 16)} = 6.541$, $p = 0.0211$, effect of Alcohol: $F_{(1, 16)} = 5.84$, $p = 0.0280$, Effect of Rapamycin: $F_{(1, 16)} = 5.224$, $p = 0.0363$; water and alcohol within the vehicle group, $p = 0.0137$, vehicle and rapamycin within the alcohol group, $p = 0.0165$. (**D**) Alcohol x Rapa-mycin: $F_{(1, 16)} = 2.198$, $p = 0.1576$, effect of Alcohol: $F_{(1, 16)} = 0.1552$, $p = 0.6989$, Effect of Rapamycin: $F_{(1, 16)} = 0.0019$, $p = 0.9656$. (**E–H**) A seperate cohort of mice underwent 7 weeks of IA20%2BC or water only. The NAc was dissected and Trax (**E**, **G**) and GW182 (**F**, **H**) protein levels were determined by western blot analysis. ImageJ was used for optical density quantification. Data are presented as the average ratio of Trax or GW182 to Tubulin±SEM and are expressed as % of water control. ***$p < 0.001$. $n = 8$ mice per group. Significance was determined using two-tailed unpaired t-tests. (**G**) $t_{(14)} = 4.354$, $p = 0.0007$; (**H**) $t_{(14)} = 4.354$, $p = 0.0007$. (**I**) Alcohol activates mTORC1 signaling in the NAc which in turn increases the trans-lation of GW182 and Trax. Source data are provided as a Source Data file.

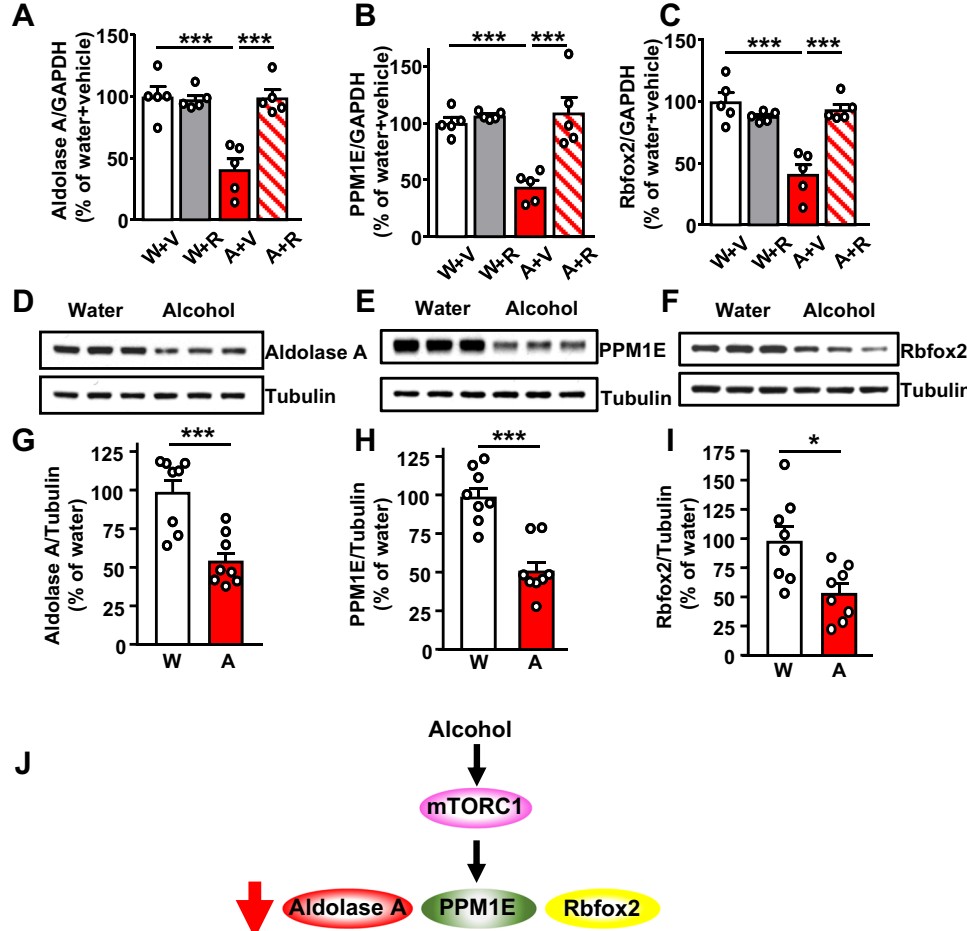

**Fig. 2 | Alcohol via mTORC1 represses the translation of Aldolase A, PPM1E and Rbfox2 in the NAc.** Mice underwent 7 weeks of IA20%-2BC (Supplementary Table 1) and were treated with 20 mg/kg rapamycin as described above. **A–C** Polysomal mRNA levels of Aldolase A (**A**), PPM1E (**B**) and Rbfox2 (**C**). Each data point represents an average of 3 technical replicates. Data are presented as the average ratio of each transcript to GAPDH ± SEM and expressed as the % of water +vehicle. ***$p < 0.001$. $n = 5$ mice per group. Significance was determined using Two-way ANOVA followed by Tukey's multiple comparisons test. **A** Alcohol x Rapamycin: $F_{(1, 16)} = 18.01$, $p = 0.0006$, effect of Alcohol: $F_{(1, 16)} = 16.38$, $p = 0.0009$, Effect of Rapamycin: $F_{(1, 16)} = 15.56$, $p = 0.0012$; water and alcohol within the vehicle group, $p = 0.0001$, vehicle and rapamycin within the alcohol group, $p = 0.0001$. **B** Alcohol x Rapamycin: $F_{(1, 16)} = 12.86$, $p = 0.0025$, effect of Alcohol: $F_{(1, 16)} = 10.51$, $p = 0.0051$, Effect of Rapamycin: $F_{(1, 16)} = 19.13$, $p = 0.0005$;

water and alcohol within the vehicle group, $p = 0.001$, vehicle and rapamycin within the alcohol group, $p = 0.0002$. **C** Alcohol x Rapamycin: $F_{(1, 16)} = 27.39$, $p < 0.0001$, effect of Alcohol: $F_{(1, 16)} = 18.9$, $p = 0.0005$, Effect of Rapamycin: $F_{(1, 16)} = 11$, $p = 0.0044$; water and alcohol within the vehicle group, $p < 0.0001$, vehicle and rapamycin within the alcohol group, $p < 0.0001$. **D–I** Aldolase A (**D, G**), PPM1E (**E, H**) and Rbfox2 (**F, I**) protein levels were determined by western blot analysis. Data are presented as the average ratio of Aldolase A, PPM1E and Rbfox2 to Tubulin±SEM and are expressed as the % of water control. *$p < 0.05$, ***$p < 0.001$. $n = 8$ mice per group. Significance was determined using two-tailed unpaired t-tests. **G** $t_{(14)} = 4.413$, $p = 0.0006$; **H** $t_{(14)} = 5.426$, $p < 0.0001$; **I** $t_{(14)} = 2.917$, $p = 0.0113$. **J** Alcohol activates mTORC1 signaling in the NAc which in turn decreases the translation of Aldolase A, Rbfox2, PPM1E. Source data are provided as a Source Data file.

As mTORC1 is downstream of H-Ras/PI3K/AKT pathway[13,14], we tested whether chronic excessive alcohol intake activates mTORC1 in the NAc of rodents, and found that alcohol drinking produces a robust and long-lasting activation of mTORC1 signaling in the NAc[15] and specifically in NAc shell[16]. Using the selective mTORC1 inhibitor, rapamycin[17], we showed that mTORC1 in the NAc contributes to the development and/or maintenance of excessive alcohol use[15,18–22]. To elucidate the mechanism(s) by which mTORC1 in the NAc drives excessive alcohol use, we conducted an RNAseq study and found that alcohol-mediated mTORC1 activation in the NAc of male mice led to an increase in the translation of 12 transcripts[19]. Among the identified transcripts was the postsynaptic protein, Prosapip1[19]. We found that the translation of Prosapip1 is increased in response to alcohol-mediated mTORC1 activation, which in turn promotes the formation of F-actin, leading to synaptic and structural plasticity, alcohol self-administration and reward[19].

Our RNAseq data also suggested that alcohol-mediated activation of mTORC1 in the NAc increases the translation of the microRNA machinery transcripts[19]. The microRNA machinery is responsible for the repression of translation and mRNA degradation via the action of ~22-nucleotide sequences, microRNAs[23,24]. After transcription by RNA polymerase II, pri-microRNAs are processed within the nucleus into pre-miRNAs by the microprocessor complex composed of Drosha and DGCR8 proteins[25]. Once exported into the cytosol, the ribonuclease Dicer produces mature miRNAs that are then loaded into the miRNA-Induced Silencing Complex (miRISC)[25]. A perfect complementarity between a miRNA and its target mRNA sequence leads to the cleavage of the target mRNA by miRISC, resulting in mRNA degradation[26]. An imperfect complementarity, however, results in reduced translation and/or stability of target mRNAs[26].

Here, we show that alcohol consumption activates mTORC1 in male NAc D1 neurons, leading to increased translation of miR machinery components and miRs expression, including miR-34a-5p,

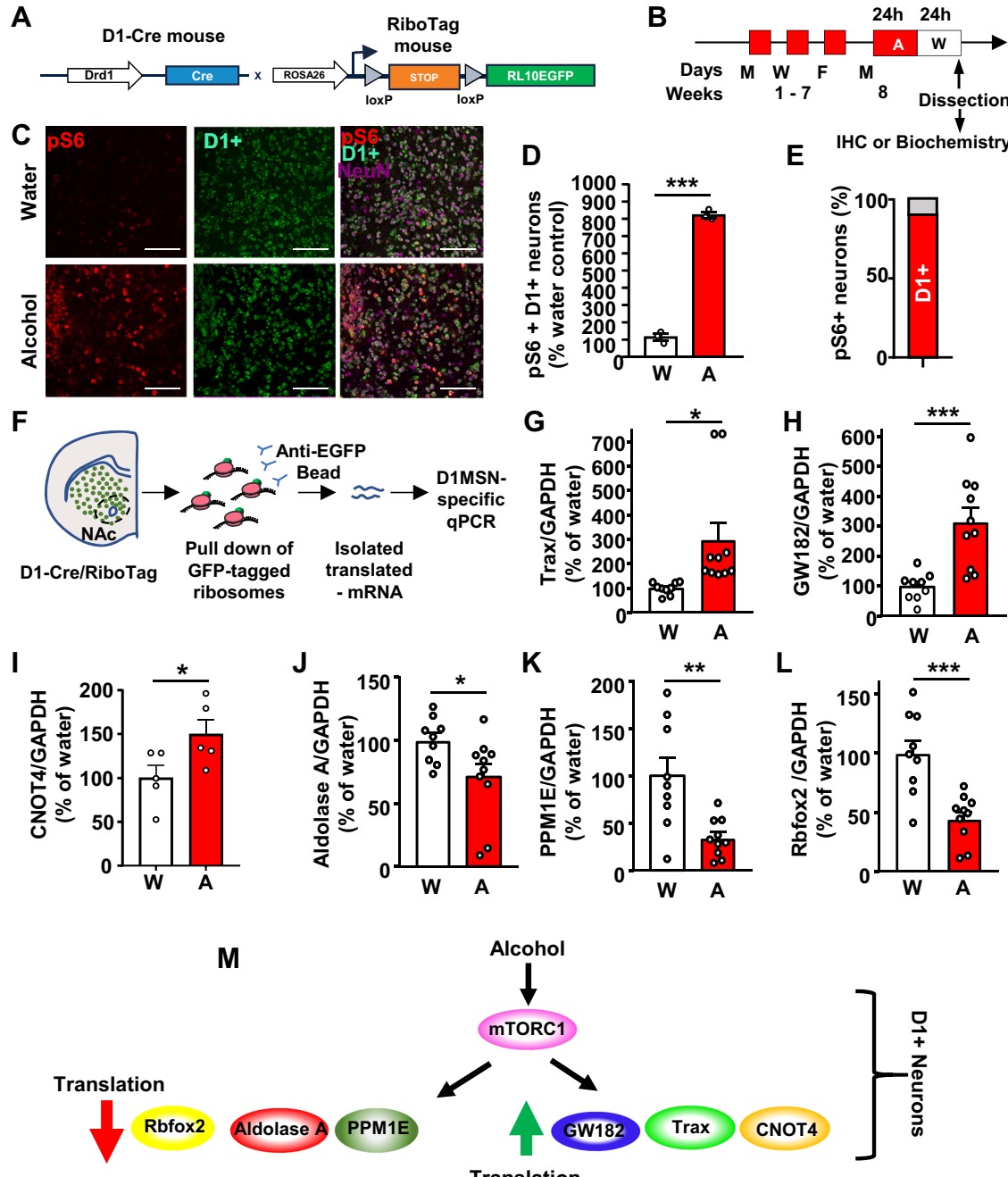

**Fig. 3 | Alcohol activates mTORC1, increases the translation of Trax, GW182 and CNOT, and decreases the translation of Aldolase A, PPM1E, Rbfox2 in NAc D1+ neurons. A** D1-Cre mice were crossed with RiboTag mice allowing the expression of RPL10-EGFP in D1-expressing neurons. **B** Mice underwent 7 weeks of IA20%2BC (Supplementary Table 1). Brains were dissected at the end of the last 24-hour withdrawal session as in Fig. 1A and processed for IHC or biochemical analysis. **C** IHC analysis of phospho-S6 levels in the NAc shell D1 neurons of drinking mice compared to water controls. Representative images of NAc at 20x magnification labeled with phospho-S6 in red, RPL10-GFP in green and NeuN in magenta. Scale bar 100 μm. **D** Phospho-S6 and D1 labeled neurons are expressed as % of water controls±SEM. ***$p$ < 0.001. $n$ = 3 mice per group. Significance was determined using two-tailed unpaired t-tests. t(4) = 27.29, $p$ = <0.0001. **E** Percentage of D1 positive vs. D1 negative NAc shell neurons labeled as phosphoS6 positive neurons. **F** Affinity purification of ribosomes from D1+ neurons by using anti-GFP magnetic

beads followed by RNA isolation and RT-qPCR. Image provided by Servier Medical Art (https://smart.servier.com/), licensed under CC BY 4.0 (https://creativecommons.org/licenses/by/4.0/). **G–L** Polysomal mRNA levels in D1+ neurons of Trax (**G**), GW182 (**H**), CNOT4 (**I**), Aldolase A (**J**), PPM1E (**K**), Rbfox2 (**L**) after alcohol withdrawal were determined by RT-qPCR. Each data point represents an average of 3 technical replicates. Data are presented as the average ratio of a transcript to GAPDH ± SEM and expressed as % of water control. *$p$ < 0.05, **$p$ < 0.01, ***$p$ < 0.001. **G, H, J–L** Water: $n$ = 9, Alcohol: $n$ = 10, **I** $n$ = 5 mice per group. Significance was determined using two-tailed unpaired t-tests. **G** t(17) = 4.042, $p$ = 0.0008; **H** t(17) = 2.575, $p$ = 0.0197; **I** t(8) = 2.322, $p$ = 0.0488; **J** t(17) = 2.299, $p$ = 0.0345; **K** t(17) = 3.425, $p$ = 0.0032; **L** t(17) = 4.26, $p$ = 0.0005. **M** Alcohol activates mTORC1 signaling in D1+ NAc neurons which in turn increases the translation of GW182, Trax and CNOT4 and represses the translation of Aldolase A, Rbfox2 and PPM1E. Source data are provided as a Source Data file.

which in turn promotes a paradoxical repression in translation of transcripts including Aldolase A. The miR-34a-5p↑/Aldolase A↓ axis in D1 NAc neurons attenuates glycolysis and promotes further drinking.

## Results

### Alcohol increases the translation of Trax and GW182 in an mTORC1-dependent manner in the NAc

As mentioned above, using RNAseq analysis we previously identified the microRNA machinery components Trax, GW182 and CNOT4 as candidate transcripts whose translation was increased by alcohol in an mTORC1-dependent manner[19]. To confirm the RNAseq data, male mice underwent intermittent access to 20% alcohol in a 2-bottle choice (IA20%2BC) paradigm for 7 weeks. This paradigm models humans who exhibit alcohol use disorder (AUD)[27]. Three hours before the end of the last 24 h alcohol withdrawal session, mice were systemically administered with vehicle or the selective mTORC1 inhibitor, rapamycin (20 mg/kg). The NAc was removed 24 h after the last alcohol withdrawal session, polysomes were purified to isolate actively translating mRNAs, and RT-qPCR analysis was conducted (Fig. 1A, Supplementary Tables 1- and 2, **alcohol drinking and statistical analysis**). First, we replicated the published data[19] in a new cohort of animals and confirmed that the translating levels of Trax were indeed increased by alcohol and reduced back to baseline when mice were first pretreated with the mTORC1 inhibitor, rapamycin (Fig. 1B). Next we confirmed the RNAseq data using RT-qPCR and showed that the translation of GW182 is increased by alcohol in the NAc in an mTORC1-dependent manner (Fig. 1C). In contrast, the RNAseq data of CNOT4 in total NAc were not replicated (Fig. 1D). Since GAPDH is used as an internal control, we analyzed its level in response to alcohol and found no change in its translation in response to alcohol (Supplementary Fig. 1).

Next, a new cohort of mice that underwent 7 weeks of IA20%2BC was used to measure the total mRNA of GW182 and Trax in alcohol and water drinking mice. We did not observe a change in the total amount of mRNA of either transcript (Supplementary Fig. 2A, B, Supplementary Tables 1-2). Together, these data suggest that alcohol increases the translation but not the transcription and/or RNA stability of the microRNA machinery components GW182 and Trax.

Finally, we analyzed the protein levels of Trax and GW182 in water and alcohol drinking mice and found that the levels of both proteins were increased in the NAc of mice consuming alcohol as compared to water only drinking mice (Fig. 1E–H, Supplementary Tables 1-2). This increase was specific to the NAc as the levels of Trax and GW182 were not altered by alcohol in a neighboring striatal region, the dorsolateral striatum (DLS) (Supplementary Fig. 2C–F, Supplementary Tables 1-2). Together, these data suggest that chronic heavy alcohol use increases the levels of the microRNA machinery proteins Trax and GW182 in the NAc of drinking mice (Fig. 1I).

### Alcohol represses the translation of PPM1E, Aldolase A and Rbfox2 in an mTORC1-dependent manner in the NAc

The microRNA machinery is responsible for repression of translation or mRNA degradation[23,24]. We speculated that the alcohol-mediated increase in the translation of Trax and GW182 may point towards an unexpected link between mTORC1 and the microRNA machinery. Interestingly, while analyzing the RNAseq data[19], we observed that the translation of 32 transcripts was repressed by alcohol in an mTORC1-dependent manner (Supplementary Table 3). We classified the transcripts and found that 37% of them belong to the signal transduction category, 31% are part of the DNA/RNA machinery, 12% transcripts are associated with actin/cytoskeleton, and 12% with metabolic pathways (Supplementary Fig. 3). We confirmed the RNAseq data of 3 random transcripts by measuring the mRNA levels in NAc polysomes of mice that underwent 7 weeks of IA20%2BC, and that were treated with vehicle or rapamycin (20 mg/kg) 3 h before the last alcohol withdrawal

session (Fig. 2A–C, Supplementary Tables 1-2). We found that the translation of the glycolytic enzyme, Aldolase A[28] (Fig. 2A), the serine/threonine Protein Phosphatase PPM1E (Mg2 + /Mn2+ Dependent 1E)[29] (Fig. 2B), and the RNA binding protein Rbfox2 (RNA binding fox 1 homolog 2)[30] (Fig. 2C) was repressed by alcohol as compared to water only drinking mice. Importantly, we detected a reversal of translation repression in mice that were pre-treated with rapamycin (Fig. 2A–C). In contrast, we did not observe a change in the total mRNA levels of the transcripts suggesting that alcohol decreases the translation but not the transcription and/or mRNA stability of these 3 transcripts (Supplementary Fig. 4A–C, Supplementary Tables 1-2).

Next, we analyzed the protein level of Aldolase A, PPM1E and Rbfox2 in water and alcohol drinking mice. We found that alcohol significantly decreases the level of the 3 proteins in the NAc (Fig. 2D–I, Supplementary Tables 1-2). The decrease was specific to the NAc, as the level of Aldolase A, PPM1E and Rbfox2 was not altered by alcohol in the DLS (Supplementary Fig. 4D–I, Supplementary Tables 1-2). Together, these data suggest that chronic heavy alcohol drinking increases the levels of the microRNA machinery proteins Trax and GW182 in the NAc and concomitantly represses the translation of Aldolase A, PPM1E and Rbfox2 (Fig. 2J).

### Alcohol-mediated mTORC1-dependent increase and decrease in translation is localized to D1 NAc neurons

The NAc is composed mainly of two neuronal subpopulations, dopamine D1 receptor (D1) and dopamine D2 receptor (D2) expressing medium spiny neurons (MSN)[31]. We previously found that mTORC1 is specifically activated by alcohol in the shell but not the core of the NAc[16] and that the first drink of alcohol activates mTORC1 specifically in D1 neurons in the NAc shell[20]. First, to determine if heavy chronic alcohol use activates mTORC1 in D1 NAc MSN, male D1-Cre mice were crossed with RiboTag mice in which GFP-fused ribosomal subunit RPL10 is expressed in the presence of Cre recombinase, enabling the detection of D1 neurons[32] (Fig. 3A). D1-Cre x RiboTag mice underwent 7 weeks of IA20%2BC and were sacrificed at the end of the last alcohol withdrawal session (Fig. 3B, Supplementary Tables 1-2). We found that alcohol significantly increases S6 phosphorylation in D1 neurons of the NAc shell (Fig. 3C, D) and that only 8.9% of phosphoS6 positive neurons are not D1 neurons (Fig. 3E). Together, these data suggest that the majority of mTORC1 activated by excessive alcohol use is localized to D1 NAc neurons.

Next, we examined whether mTORC1-dependent alterations in translation are also localized to D1 NAc MSNs. To do so, we utilized D1-Cre x RiboTag mice which contain a GFP tag for affinity purification of D1 neuron ribosomes allowing for the isolation of mRNAs actively undergoing translation in D1 MSNs (Fig. 3F). D1-Cre x RiboTag mice underwent 7 weeks of IA20%2BC or water only, and mRNA levels were measured in the GFP immunoprecipitated fraction (Fig. 3F, Supplementary Tables 1-2). We detected an enrichment in Trax and GW182 translation in D1 NAc neurons (Fig. 3G, H). Interestingly, we also detected an increase in the level of CNOT4 mRNA undergoing translation in D1 MSNs (Fig. 3I). In parallel, translation of Aldolase A, PPM1E, and Rbfox2 was attenuated by alcohol in NAc D1 MSNs (Fig. 3J–L). To test whether these increases and decreases in translation were specific to D1 NAc neurons, we utilized A2A-Cre (Cre expressed in D2 neurons) x RiboTag mice. mRNA levels were measured in the GFP immunoprecipitated fraction in water or alcohol drinking A2A-CrexRiboTag mice. We did not detect an increase in Trax, GW182 and CNOT translation in D2 NAc neurons (Supplementary Fig. 5A–C). In addition, PPM1E, Rbfox2, and Aldolase A levels were not decreased in D2 neurons (Supplementary Fig. 5D–F). Together, our data suggest that mTORC1-dependent translation activation and translation repression occur specifically in NAc D1 neurons (Fig. 3M).

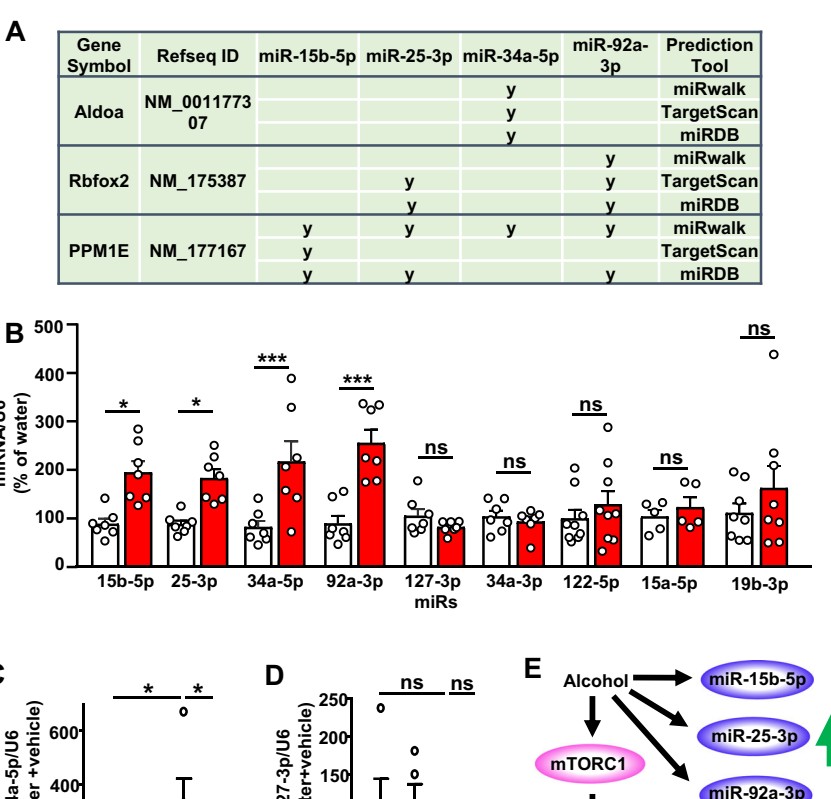

**Fig. 4 | Identification of miR-15b-5p, miR-25-3p, miR-92-3p and miR-34a-5p which are increased by alcohol in the NAc; Activation of mTORC1 is required for alcohol-mediated increase of miR-34a-5p expression. A** Potential miRNA-target interaction between miR-15b-5p, miR-25-3p, miR-92a-3p and miR-34a-5p the transcripts of interest: PPM1E, Aldolase A and Rbfox2. miRNA-target interactions were determined using miRWalk, TargetScan and miRDB. **B** Mice underwent 7 weeks of IA20%2BC or water only (Supplementary Table 1). Three hours before the end of the last 24 h of alcohol withdrawal, the NAc was removed and the expression of miR-15b-5p, miR-25-3p, miR-34a-5p and miR-92-3p, as well as miR-127-3p and miR-34a-3p, miR-122-5p, miR-15a-5p and miR-19b-3p were measured by RT-qPCR. Each data point represents an average of 3 technical replicates. Data are presented as individual data points and mean ± SEM. *$p < 0.05$, ***$p < 0.001$, ns: non-significant. $n = 7$ mice per group. Significance was determined using Two-way ANOVA followed by Sidak's multiple comparisons test. Alcohol x miR: F (8, 110) = 3.994, $p = 0.0003$, effect of Alcohol: F(1, 110) = 33.14, $p < 0.0001$, Effect of miR: F(8,110) = 2.428, $p = 0.0187$; miR15b-5p water vs. alcohol, $p = 0.0147$; miR 25-3p water vs. alcohol, $p = 0.0402$; miR 34a-5p water vs. alcohol, $p = 0.0007$; miR 92a-3p water vs. alcohol, $p < 0.0001$; miR 127-3p water vs. alcohol, $p = 0.9977$; miR 34a-3p water vs. alcohol, $p > 0.9999$; miR 122-5p water vs. alcohol, $p = 0.9707$; miR

15a-5p water vs. alcohol, $p = 0.9998$; miR 19b-3p water vs. alcohol, $p = 0.6109$. **C, D** Mice underwent 7 weeks of IA20%2BC or water only (Supplementary Table 1). Three hours before the end of the last 24 h of alcohol withdrawal session, mice that consumed alcohol (A) or water only (W) were injected with rapamycin (20 mg/kg) (R) or vehicle (V). The levels of miR-34a-5p (**C**) and miR-127-3p (**D**) were measured by RT-qPCR. Each data point represents an average of 3 technical replicates. Data are presented as individual data points and mean ± SEM. *$p < 0.05$, ns: non-significant. Water+Vehicle: $n = 5$ mice, Water+Rapamycin: $n = 5$ mice, Alcohol+vehicle: $n = 5$ mice, Alcohol+Rapamycin: $n = 6$ mice. Significance was determined using Two-way ANOVA followed by Sidak's multiple comparisons test. **C** Alcohol x Rapamycin: F(1, 17) = 7.291, $p = 0.0152$, effect of Alcohol: F(1, 17) = 5.107, $p = 0.0372$, Effect of Rapamycin: F(1, 17) = 2.621, $p = 0.1238$; water and alcohol within the vehicle group, $p = 0.0189$, vehicle and rapamycin within the alcohol group, $p = 0.0366$. **D** Alcohol x Rapamycin: F(1, 17) = 0.01218, $p = 0.9134$, effect of Alcohol: F(1, 17) = 2.866, $p = 0.1087$, Effect of Rapamycin: F(1, 17) = 0.002773, $p = 0.9586$. **E** Alcohol increases the levels of miR-15b-5p, miR-25-3p, miR-92-3p and miR-34a-5p which are predicted to target the 3 transcripts shown in Fig. 3. Alcohol-mediated miR-34a-5p increase depends on mTORC1. Source data are provided as a Source Data file.

## Identification of microRNAs targeting PPM1E, Aldolase A and Rbfox2

To determine whether miRs are responsible for the translation repression of PPM1E, Aldolase A and Rbfox2, we performed in silico analyses using algorithmic prediction tools miRWalk, miRDB and TargetScan to identify putative microRNAs targeting the 3 repressed transcripts. miR15b-5p, miR25-3p, miR-34a-5p and miR92a-3p were identified by all 3 prediction tools as potential specific miRs for PPM1E, Aldolase A, and Rbfox2 (Fig. 4A). We examine the level of the predicted miR in the NAc. Interestingly, we found that the level of all 4 miRs was elevated by alcohol in the NAc (Fig. 4B, Supplementary Tables 1-2). However, the expression of other miRs; miR127-3p, miR122-5p, miR19b-3p, miR15a-5p, and miR-34a-3p were not altered

by alcohol (Fig. 4B, Supplementary Tables 1-2). Since U6 was used as an internal control, we analyzed its level in response to alcohol and found no change (Supplementary Fig. 6). The alcohol-mediated increase in miR levels was specific to the NAc since the levels of these miRs were not altered by alcohol in the DLS (Supplementary Fig. 7, Supplementary Tables 1-2). Together, these data suggest the expression of miRs that are predicted to target Aldolase A, PPM2E and RbFox2 are increased in the NAc of mice that drink alcohol.

## miR-34a-5p targets Aldolase A in the NAc

We next focused on Aldolase A and its predicted miR, miR-34a-5p. First, we examined if the alcohol-mediated increase in miR-34a-5p expression in the NAc depends on mTORC1 and found that alcohol-

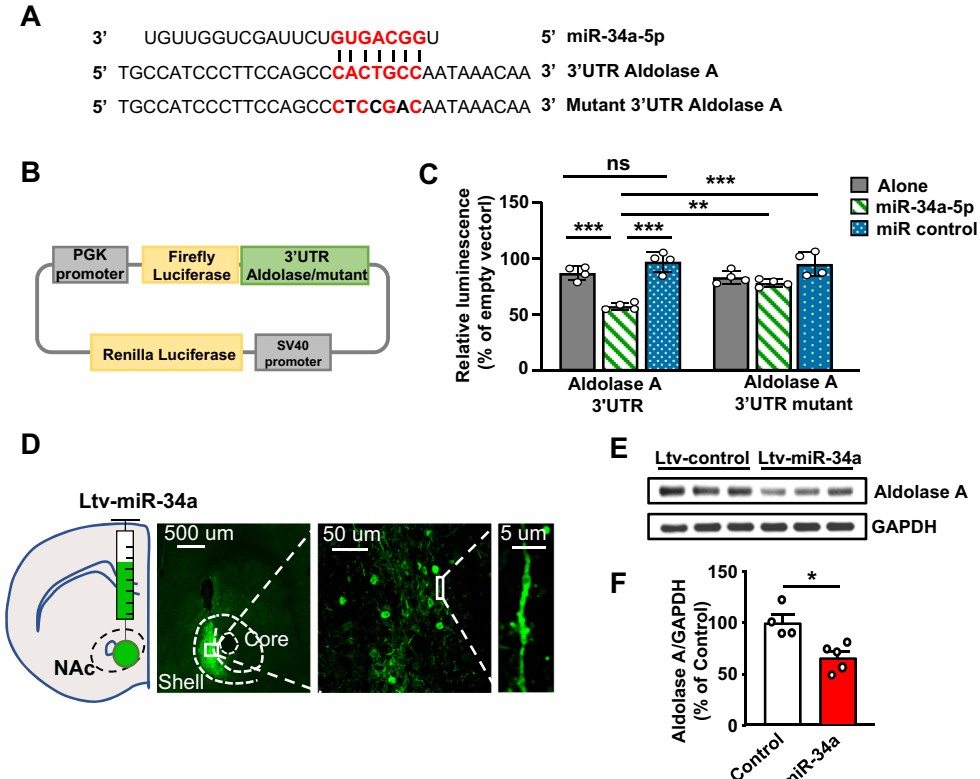

**Fig. 5 | miR-34a-5p interacts with Aldolase A 3'UTR and represses Aldolase A levels in the NAc. A** Predicted interaction sites (in red) of miR-34a-5p sequence (top) and Aldolase A 3'UTR (middle). Mutated Aldolase 3'UTR is shown at the bottom. **B** Map of Aldolase A 3'UTR or mutant 3'UTR cloned into a luciferase reporter vector. **C** HEK293 cells were co-transfected with a reporter vector containing Aldolase A 3'UTR or mutant 3'UTR and miR-34a-5p or a negative control. Bar graph depicts average ± SD expressed as Firefly luminescence normalized to Renilla luminescence and relative to a reference control. Gray bars: Aldolase A or mutant Aldolase A. Hatched green bars: Aldolase A or mutant Aldolase A + miR-34a5p. Dotted blue bars: Aldolase A or mutant Aldolase A + miR control. ** $p < 0.01$, *** $p < 0.001$, ns: non-significant. $n = 4$ independent experiments per group. Each data point represents an average of 3 technical replicates. Significance was determined using Two-way ANOVA followed by Tukey's multiple comparisons test. miR x Aldolase A 3'UTR: $F(2, 18) = 7.882$, $p = 0.0035$, $p = 0.0003$, effect of Aldolase A 3'UTR: $F(1, 18) = 3.144$, $p = 0.0931$, Effect of miR: $F(2, 18) = 34.75$, $p < 0.0001$; Aldolase 3'UTR + miR34a-5p vs. Aldolase A 3'UTR + miR control, $p < 0.0001$; Aldolase A 3'UTR mutant + miR34a-5p vs. Aldolase A 3'UTR + miR34a-5p, $p = 0.0053$; Aldolase A 3'UTR + miR34a-5p vs. Aldolase A 3'UTR mutant + miR control, $p < 0.0001$. **D** Lenti-miR-34a-GFP infected NAc neurons. **E, F** miR-34a-5p or GFP was expressed in the NAc and Aldolase A protein level was evaluated by western blot analysis, quantified as a ratio of Aldolase A/GAPDH ± SEM and depicted as % of Aldolase A levels in the NAc of GFP infected mice. * $p < 0.05$. Water: $n = 4$ mice, Alcohol: $n = 5$ mice. Significance was determined using two-tailed Mann-Whitney test. **F** U = 0, $p = 0.0159$. Source data are provided as a Source Data file.

mediated increase in miR-34a-5p expression is attenuated in mice that were pretreated with rapamycin (Fig. 4C, Supplementary Tables 1-2). In contrast, the level of miR127-3p was unaltered in mice drinking alcohol and treated with rapamycin (Fig. 4D, Supplementary Tables 1-2). We hypothesized that miR-34a-5p is required for the repression of Aldolase A translation and speculated that miR-34a-5p binds 3'UTR of Aldolase A. To test this possibility, we constructed luciferase expression plasmids containing Aldolase A 3'-UTR sequence or a mutant of the 3'UTR which does not contain the putative miR-34a-5p interaction site (Fig. 5A, B). Co-transfection of the miR-34a-5p with Aldolase A 3'-UTR plasmids into HEK293 cells significantly suppressed luciferase activity, while co-transfection of Aldolase A 3'-UTR with a control miR did not affect the luciferase activity (Fig. 5C, Supplementary Table 2). Mutation of the miR binding site within the Aldolase A 3'-UTR sequence abolished the effect of miR-34a-5p on the luciferase activity (Fig. 5C, Supplementary Table 2). These data demonstrate that Aldolase A 3'-UTR directly interacts with miR-34a-5p. To test whether miR-34a-5p is targeting Aldolase A in vivo, the NAc shell of mice was infected with a lentivirus expressing miR-34a (Ltv-miR-34a) (Fig. 5D). As shown in Fig. 5E, F (Supplementary Table 2), Aldolase A levels in the NAc were significantly reduced by miR-34a overexpression. Together, these data suggest that miR-34a is increased by alcohol via mTORC1 which in turn targets Aldolase A for translation repression.

## Alcohol reduces glycolysis in the NAc via mTORC1

Aldolase A belongs to the family of the Fructose Diphosphate Aldolase enzymes[28]. Aldolase A is highly expressed in neurons and is the major isoform expressed in the striatum[33,34]. Aldolase C is also expressed in the brain[35], however we found that Aldolase C protein levels in the NAc were unaltered by alcohol (Supplementary Fig. 8). Aldolase A is a critical enzyme in glycolysis, a ten-step metabolic pathway resulting in the production of lactate[36], and ATP molecules through the tricarboxylic acid (TCA) cycle in the mitochondria[37]. Aldolase A catalyzes the conversion of Fructose 1,6-Bisphosphate (F1,6BP) to Glyceraldehyde 3-phosphate (G3P) and dihydroxyacetone phosphate (DHAP)[37] (Fig. 6A). Since alcohol represses Aldolase A translation and protein levels in the NAc, we hypothesized that alcohol-mediated activation of mTORC1 in D1 NAc neurons leads to a reduction in glycolysis. To examine the possibility, we first conducted a metabolomics study in which we used mice that underwent IA20%2BC for 7 weeks (Supplementary Table 1). Twenty-four h after the last drinking session, the NAc was dissected followed by metabolites extraction and mass spectrometry analysis. We found that the end product of glycolysis, lactate[36], as well as other metabolites within the TCA cycle such as citrate, a-ketoglutarate and malate, were reduced by alcohol (Fig. 6B, Supplementary Table 2), suggesting that alcohol reduces glycolysis in the NAc. The attenuation of glycolysis was also not due to changes in

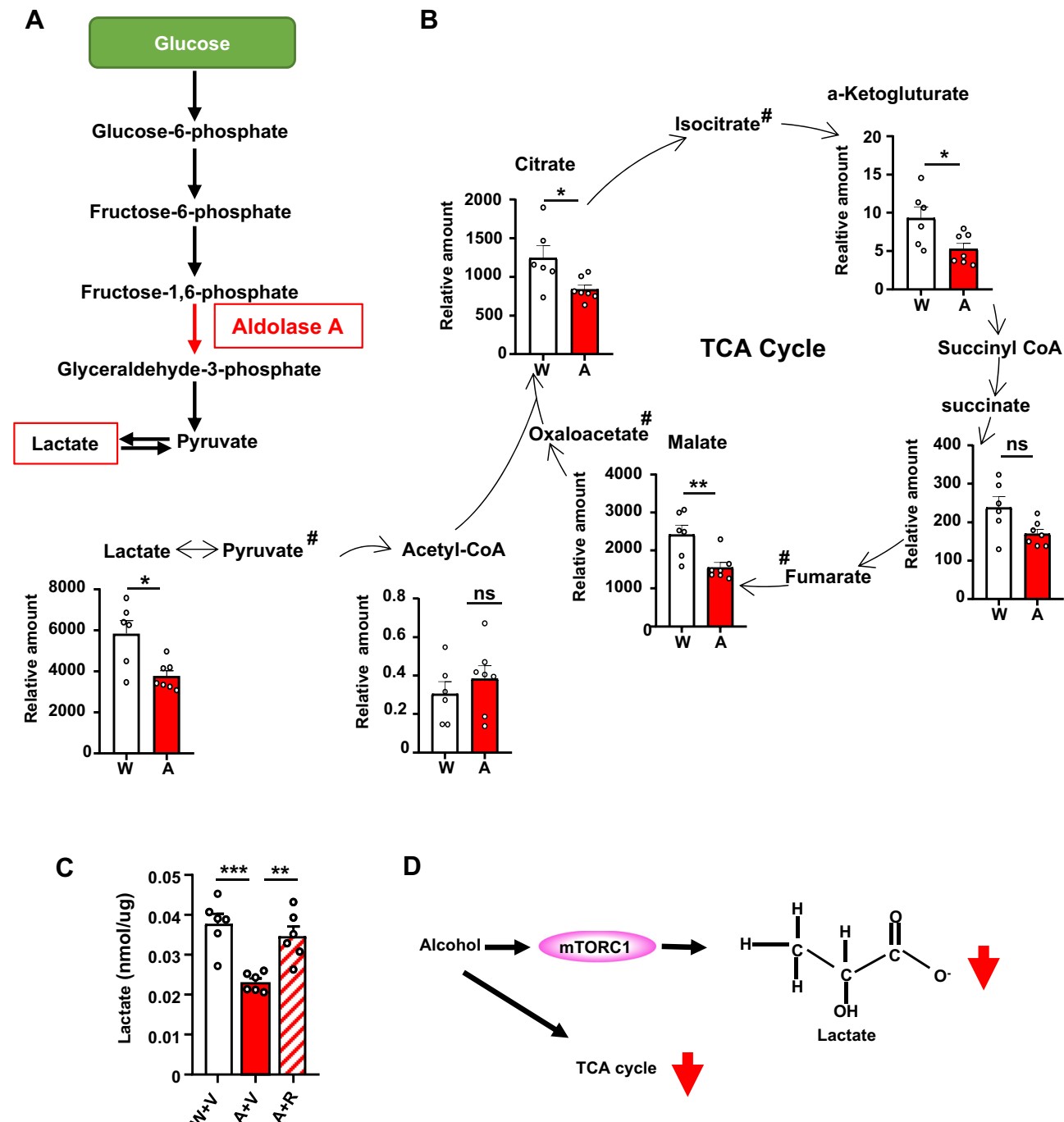

**Fig. 6 | Alcohol decreases TCA cycle metabolites, including lactate which depends on mTORC1. A** Aldolase A in the glycolysis pathway converts F1,6BP to G3P and DHAP. Lactate is the final product of glycolysis. **B** After 7 weeks of IA20%2BC (Supplementary Table 1) or water only, the NAc was dissected after 24 h of alcohol withdrawal and metabolite levels were measured. Data are presented as relative amount of individual metabolites in water vs. alcohol. Data are presented as individual data points and mean ± SEM. *$p < 0.05$, **$p < 0.01$, ns: non-significant. # metabolites that were not included in the panel. Water control: $n = 6$ mice, Alcohol withdrawal: $n = 7$ mice. Significance was determined using two-tailed Mann-Whitney test. Lactate U = 4, $p = 0.0140$; Citrate U = 6, $p = 0.035$; a-Ketogluturate U = 6, $p = 0.035$; Malate U = 3, $p = 0.0082$. **C** Mice underwent 7 weeks of IA20%2BC (Supplementary Table 1) or water only. Three hours before the end of the last alcohol withdrawal period, mice were systemically injected with 20 mg/kg rapamycin or vehicle. The NAc was dissected after 3 h, and lactate level was measured using a colorimetric assay. Data are presented as individual data points and mean ± SEM. Each data point represents an average of 3 technical replicates. **$p < 0.01$, ***$p < 0.001$. $n = 6$ mice per group. Significance was determined using One-way ANOVA followed by Tukey's multiple comparisons test. F(2, 15) = 13.48, $p = 0.0004$. Water/vehicle vs alcohol/vehicle, $p = 0.0005$; alcohol/vehicle vs alcohol/rapamycin, $p = 0.004$. **D** Alcohol reduces TCA metabolites in the NAc and lactate which depends on mTORC1. Source data are provided as a Source Data file.

the expression of the main neuronal and astrocyte glucose or lactate transporters (Supplementary Fig. 9)[19]. Ter Horst reported that D1 MSNs in the NAc of mice regulate glucose tolerance sensitivity in the periphery[38]. We therefore investigated whether glucose tolerance was affected in alcohol drinking mice. As shown in Supplementary Fig. 10, blood glucose level during the glucose tolerance test was not altered by 4 or 7 weeks of IA20%2BC, suggesting that alcohol's effect on glucose metabolism is centrally localized.

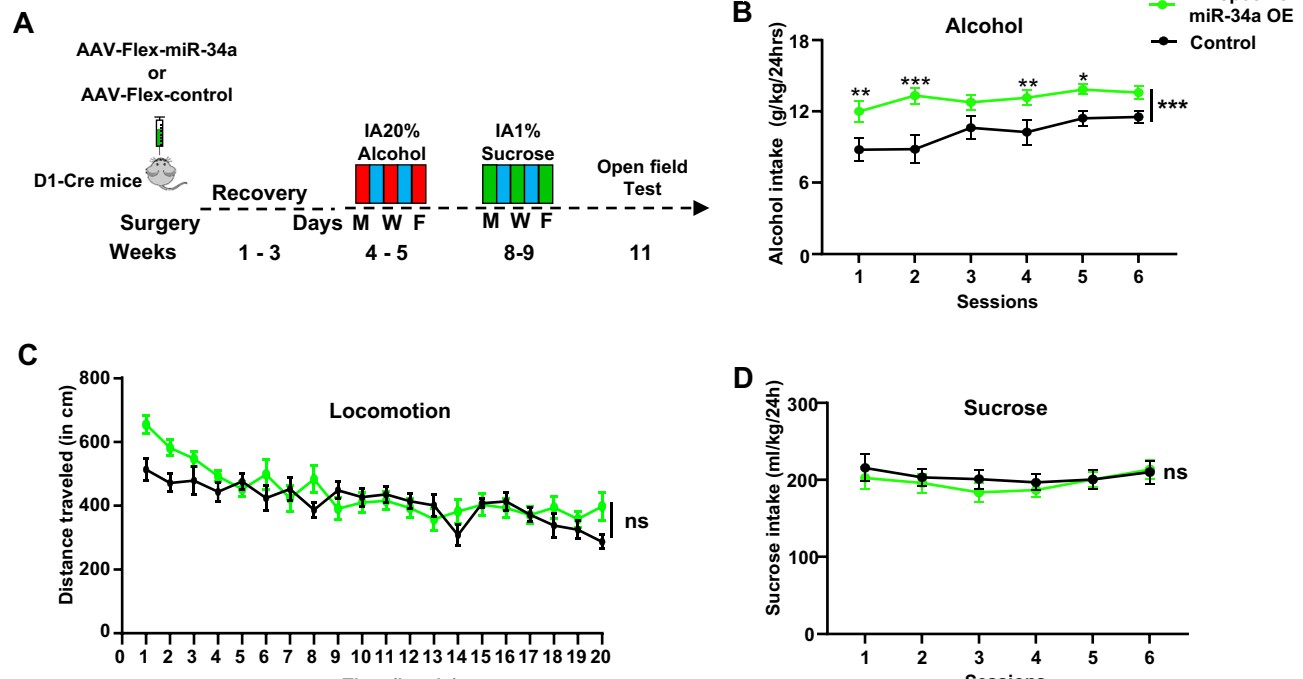

**Fig. 7 | Overexpression of miR-34a in NAc D1 neurons increases alcohol consumption but does not affect locomotion or sucrose intake. A** Experimental timeline. Mice received a bilateral infusion of AAV2-Flex-miR-34a or AAV2-Flex-control in the NAc shell of D1-Cre mice. Three weeks after surgery, mice underwent 2 weeks of IA20%2BC followed by 2 weeks of water only and 2 weeks of IA1% Sucrose2BC. Mice were then subjected to the open field test. **B** The NAc of D1-Cre mice was infected with AAV2-Flex-control or AAV2-Flex-miR-34a-GFP. Three weeks later, mice underwent IA20%2BC, and alcohol consumption was measured daily for 6 sessions. Data are presented as mean ± SEM. *$p < 0.05$, **$p < 0.01$, ***$p < 0.001$. Control group: $n = 10$ mice, D1-specific miR-34a OE group: $n = 9$ mice. Significance was determined using Two-way ANOVA Mixed-effects followed by Sidak's multiple comparisons test. miR34a-5p overexpression x Time: $F_{(5, 84)} = 1.036$, $p = 0.4017$; miR34a-5p overexpression: $F_{(1, 17)} = 16.46$, $p = 0.0008$; Time: $F_{(5, 84)} = 3.698$, $p = 0.0045$. Session 1, $p = 0.0037$; Session 2, $p < 0.0001$; Session 3, $p = 0.0578$;

Session 4, $p = 0.0089$; Session 5, $p = 0.0277$; Session 6, $p = 0.061$. **C** The NAc of D1-Cre mice was infected with AAV2-Flex-control or AAV2-Flex-miR-34a. Mice were placed in an open field and mice movement was recorded for 20 min. Locomotion is depicted in 1-minute bins. Data are presented as mean ± SEM. ns: non-significant. Control group: $n = 10$ mice, D1-specific miR-34a OE group: $n = 9$ mice. Significance was determined using Two-way ANOVA RM. miR34a-5p overexpression x Time: $F_{(5, 85)} = 0.3717$, $p = 0.8667$; miR34a-5p overexpression: $F_{(5, 85)} = 0.3105$, $p = 0.5846$; Time: $F_{(5, 85)} = 1.623$, $p = 0.1627$. **D** AAV2-Flex-control or AAV2-Flex-miR-34a-GFP infected mice underwent 2 weeks of IA1%Sucrose2BC. Data are presented as mean ± SEM. ns: non-significant. Control group: $n = 10$ mice, D1-specific miR-34a OE group: $n = 9$ mice. Significance was determined using Two-way ANOVA RM. miR34a-5p overexpression x Time: $F_{(19, 323)} = 2.614$, $p = 0.0003$; miR34a-5p overexpression: $F_{(1, 17)} = 1.148$, $p = 0.2990$; Time: $F_{(17, 323)} = 10.58$, $p < 0.0001$. Source data are provided as a Source Data file.

To examine if alcohol-mediated reduction in lactate content identified in the metabolomics study is mTORC1-dependent, mice underwent 7 weeks of IA20%2BC (Supplementary Table 1). Three hours before the end of the last withdrawal session, mice were treated with vehicle or rapamycin (20 mg/kg). We found that lactate content in the NAc was significantly decreased by alcohol in mice pretreated with vehicle (Fig. 6C, Supplementary Table 2). In contrast, lactate content in the NAc returned to basal levels in mice that consumed alcohol and that were pretreated with rapamycin (Fig. 6C, Supplementary Table 2). Together, these data suggest that a consequence of the mTORC1↑/miR-34a-5p↑/Aldolase A↓ pathway is the attenuation of glycolysis and the reduction in lactate content in the NAc (Fig. 6D).

**Overexpression of miR-34a-5p in NAc D1 neurons accelerates heavy alcohol intake**
We previously found that mTORC1 in the NAc plays an important role in mechanisms underlying alcohol drinking behaviors[15,18,19]. Since we found that the activation of mTORC1 in D1 NAc neurons by alcohol promotes the increase in miR34a-5p expression leading to the attenuation of Aldolase A translation in D1 neurons, we speculated that the mTORC1↑/miR-34a-5p↑/Aldolase A↓ pathway in the NAc D1 MSNs contributes to the development of excessive alcohol consumption.

We further reasoned that if this pathway contributes to neuroadaptations that drive alcohol intake, then overexpression of miR-34a-5p in NAc D1 MSNs will repress the translation of Aldolase A, and increase alcohol consumption. To test this, the NAc shell of D1-Cre mice was infected with AAV2-Flex-miR-34a or an AAV2-Flex-control virus that only contains a fluorescent reporter (Supplementary Fig. 11). Three weeks later, mice were subjected to IA20%2BC (**Timeline** Fig. 7A). We found that overexpression of miR-34a-5p, specifically in D1 MSNs, increased and accelerated alcohol drinking as compared to mice infected with AAV2-Flex-control (Fig. 7B, Supplementary Table 2) which was not due to increased locomotive activity (Fig. 7C).

Alcohol often acts distinctly from natural reward[39,40], and we previously showed that the mTORC1 signaling in the NAc is not activated in response to sucrose intake[16]. We further discovered that inhibition of mTORC1 by rapamycin does not alter sucrose intake[15]. We hypothesized that mTORC1↑/miR-34a-5p↑/Aldolase A↓ pathway in D1 MSNs does not play a role in sucrose consumption. To test this hypothesis, D1-Cre mice infected with AAV2-Flex-miR-34a or AAV2-Flex-control were subjected to IA 1% sucrose 2BC (**Timeline** Fig. 7A). As shown in Fig. 7D, the effect of D1-specific overexpression of miR-34a was specific for alcohol as it did not affect sucrose consumption.

### L-Lactate reduces heavy alcohol intake

As depicted above the activation of mTORC1↑/miR34a-5p↑/ Aldolase A↓ pathway results in a decrease in glycolysis and its final product lactate. We speculated that replenishing lactate levels will attenuate alcohol consumption. To test this hypothesis, mice underwent 7 weeks of IA20%2BC (Supplementary Table 1) and were then treated with L-lactate (2 g/kg) supplement, which was administered subcutaneously (s.c.) 30 min before a drinking session, and alcohol and water intake were measured. We found that systemic administration of L-lactate was sufficient to reduce binge alcohol intake (Fig. 8A, Supplementary Table 2). At the 4 h time points, in parralel to the reduction of alcohol intake, a concomitent increase in water (Fig.8B) and total fluid (Fig. 8C) were detected. However, alcohol and water intake at the end of the 24-hour drinking session were not affected by L-lactate (Fig. 8D–F). Recently Lund et al. reported that the high amount of sodium in the L-lactate solution induces dehydration and significantly increases water intake[41] as detected in Fig. 7B, C, Supplementary Table 2. To rule out the possible adverse effect of sodium on alcohol drinking, a solution of sodium chloride (NaCl iso-osmolar solution, 1 g/kg) was administered 30 min before a drinking session, and alcohol and water intake were measured. NaCl-treated mice showed a significant increase in water consumption, whereas their alcohol intake was unchanged (Supplementary Fig. 12, Supplementary Table 2). These results suggest that the decrease in alcohol consumption by L-lactate administration was driven by L-lactate and not by sodium.

We next assessed whether the L-lactate effect is specific to alcohol or is shared with other rewarding substances. A new cohort of mice underwent 2 weeks of IA1%Sucrose2BC and was then treated with L-lactate (2 g/kg) subcutaneously 30 min before a drinking session. We found that L-lactate administration did not alter sucrose intake (Fig. 8G–L). Finally, we did not detect alteration in mice locomotion following the administration of L-lactate (Fig. 8M). Together, our data imply that the signaling cascade that includes miR-34a-5p and lactate drives alcohol intake (Fig. 9).

## Discussion

We found that alcohol, by activating mTORC1 in the NAc shell D1 neurons, increases the translation of microRNA machinery transcripts and the levels of miRs, including miR-34a-5p. Alcohol-dependent activation of mTORC1 also represses the translation of a number of transcripts including Aldolase A specifically in D1 NAc neurons. We further show that miR-34a-5p targets Aldolase A for translation repression. As a result of alcohol-mediated mTORC1-dependent reduction in Aldolase A levels in the NAc, glycolysis is inhibited, and lactate levels are reduced. Our results, therefore, suggest that the pathway mTORC1↑/miR-34a-5p↑/ Aldolase A↓/glycolysis↓Lactate↓ in the NAc contributes to the escalation of alcohol intake (Fig. 9).

Our study focused on male mice as mTORC1 is not activated after binge drinking or withdrawal in the NAc of female mice[42,43]. Along with these findings, we found that systemic administration of rapamycin in female mice, unlike in male mice, does not reduce alcohol consumption[43]. Similarly, Cozzoli et al. reported that intra-NAc administration of rapamycin does not change alcohol intake in female mice[42]. What could be the root cause for the sex-specific effects of alcohol on mTORC1 signaling? One possibility is that the localization of components in the mTORC1 signaling cascade in the NAc is a priori different in male vs. female mice. It is also plausible that stress[44], and/or sex hormones[45], and/or sex-chromosome components[46,47] interact with alcohol and influences the activation of mTORC1 signaling differently in the NAc of male and female mice. Together, our data suggest a sex-specific interaction between alcohol and mTORC1, which is currently being investigated.

We previously showed that both binge alcohol drinking and withdrawal activate mTORC1 in the NAc[15,16]. Here we show that mTORC1 activation, as well as translation enhancement and repression by alcohol, are localized to D1 neurons. What could be the mechanism for alcohol-mediated mTORC1 activation in NAc D1 neurons? Withdrawal from alcohol drinking increases the glutamatergic tone in the brain[48] and specifically in the NAc[49]. Stimulation of glutamate metabotropic mGlu1/5 receptors activates mTORC1 in hippocampal neurons[50], and we previously found that stimulation of the NMDA receptors (NMDARs) activates mTORC1 in the orbitofrontal cortex (OFC)[51]. Thus, it is plausible that the release of glutamate by cortical inputs specifically onto D1-MSN triggers mTORC1 activation in NAc D1 MSNs and the initiation of the cascade. We also found that mTORC1 is activated in the NAc during a short binge drinking session[15,16], a time point in which the NMDARs are inhibited[52]. Acutely, alcohol was shown to block the NMDARs in the hippocampus[53], and we previously found that acute alcohol exposure produces a robust inhibition of the activity of NMDARs in the NAc[52]. Curiously, inhibition of the NMDARs was shown to activate mTORC1 in the prefrontal cortex (PFC)[54] and in primary hippocampal neurons[55]. Furthermore, the NMDAR inhibitor and antidepressant drug, ketamine, activates mTORC1[56]. Therefore, it is plausible that during binge drinking of alcohol, mTORC1 is activated by NMDAR inhibition and that the activation is maintained via glutamate binding to the NMDAR and/or mGlu1/5 during withdrawal.

RNAseq analysis aimed to identify alcohol-mediated mTORC1-dependent translatome suggested that alcohol enhances the translation of miR machinery proteins CNOT4, Trax and GW182[19]. We found that Trax and GW182 translation was increased in the NAc and specifically in NAc shell D1 neurons of mice consuming alcohol. Interestingly, alcohol-mediated translation of CNOT4 was only detected in D1 neurons which could be due to CNOT4 signal being lost upon harvesting the whole NAc. Together, these data suggest that mTORC1 activation by alcohol upregulates the translation of Trax, GW182 and CNOT4 in D1 NAc shell neurons. GW182 is an essential part of the translation repression machinery[24,25]. Trax and its binding partner Translin were reported to promote mRNA degradation[57,58]. Chen et al. found that the mRNA silencing CCR4-NOT complex, which CNOT4 is part of, hooks onto GW182 and recruits DDX6 to repress miR-target mRNAs[59]. In parallel to the increase in the translation of the microRNA machinery transcripts, we identified alcohol-mediated mTORC1-dependent translation repression as well as the elevation of levels of specific miRs. Interestingly, hyperactivation of mTORC1 in cells results in an increase in microprocessor activity, a nuclear complex including Drosha and its partner DGCR8, thereby promoting miR biogenesis[60]. Interestingly, excessive mTORC1 activation in tuberous sclerosis complex (TSC) disorder and Alzheimer's disease leads to the reduction of specific proteins[61–63]. Further studies are required to test the direct link between the upregulation of the microRNA machinery proteins GW182, Trax, and CNOT4 and repression of translation by alcohol and whether the increase in these transcripts promotes, as we predict, the function and/or efficacy of the microRNA machinery.

Our RNAseq data suggest that one of the consequences of mTORC1 activation by alcohol in the NAc is the repression of translation of 32 transcripts. These data are unexpected as mTORC1's main role is to promote translation[4,8], whereas miRs function is to degrade mRNAs and to repress translation[23]. Previous data suggest that miRs target components of mTORC1 signaling and thus inhibit mTORC1 function[64–66]. For instance, miR199a-3p and miR100 directly target mTOR itself[64,65], and activation of mTORC1 by high level of nutrients represses miRs biosynthesis by the mRNA degradation of Drosha[66]. Thus, the fact that mTORC1 activation under certain circumstances, e.g., heavy alcohol use promotes miRNA biosynthesis and represses the translation of numerous transcripts challenges the conventional notion and suggests context-specific effects of mTORC1 signaling on miR expression.

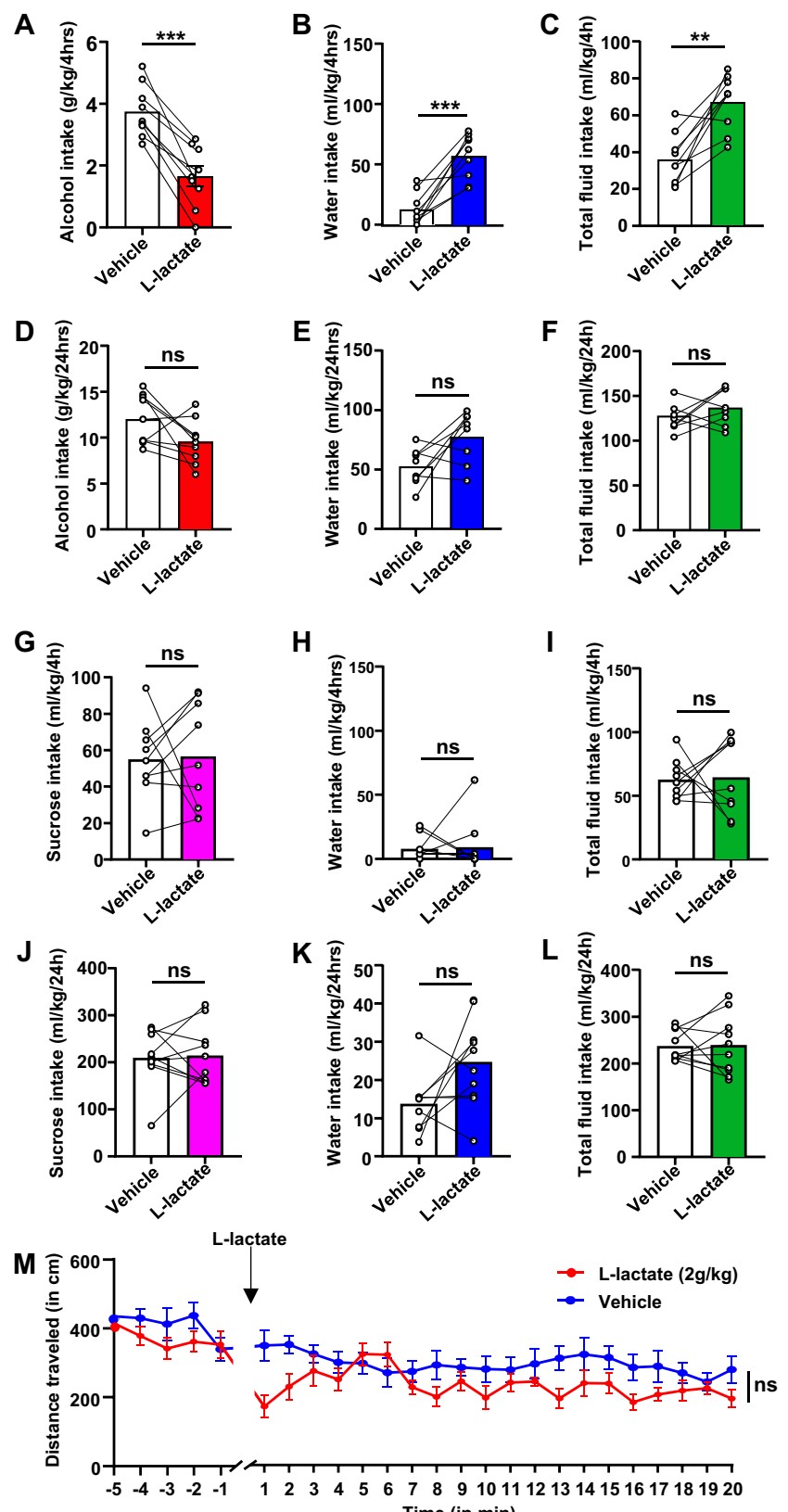

Interestingly, we observed that in parallel to the repression of translation, alcohol increases the levels of miRs that were identified by bioinformatic tools to potentially target the 3 tested transcripts. Importantly, we show that alcohol-mediated biosynthesis of at least one of the miRs (miR-34a-5p) depends on mTORC1, thus directly linking mTORC1 in the NAc to the microRNA machinery and translation repression. The miR34 family has been associated with neurodegenerative disease[67], psychiatric disorders[67–73] as well as fear, anxiety, stress[67,70,73], memory deficits and cognition[71,72]. Here, we identified new roles for miR-34a-5p that are localized to the NAc, e.g., repression of Aldolase A translation and upregulation of alcohol drinking. Interestingly, mTORC1 in TSC2-deficient cells has been shown to increase the

**Fig. 8 | Subcunenious administration of L-lactate attenuates alcohol consumption but does not affect locomotion or sucrose intake. A–F** Mice underwent 7 weeks of IA20%2BC (Supplementary Table 1). Mice received a single administration of L-lactate (s.c. 2 g/kg) or PBS on weeks 8th and 9th 30 min before the beginning of the 24 h alcohol drinking session in a counterbalanced manner. Alcohol and water were measured at the 4 h (**A–C**) and 24 h (**D–F**) time points. Data are presented as individual data points and mean ± SEM. **p < 0.01, ***p < 0.001, ns: non-significant. n = 9 mice per group. Significance was determined using two-tailed paired t-tests. **A** t(8) = 7.562, p = <0.0001; **B** t(8) = 5.917, p = 0.0004; **C** t(8) = 4.742, p = 0.0015; **D** t(8) = 2.004, p = 0.08; **E** t(8) = 2.352, p = 0.0509; **F** t(8) = 1.316, p = 0.2297. **G–L** Mice underwent 2 weeks of IA1%Sucrose2BC. Control animals had access to water only. On weeks 3 and 4, mice were s.c. injected with L-lactate (2 g/ kg) or PBS in a counterbalanced manner 30 min before the beginning of a 24-hour

drinking session. Sucrose and water intake were measured 4 h **G–I** and 24 h **J–L** later. Data are presented as individual data points and mean ± SEM. ns: non-significant. n = 9 mice per group. Significance was determined using two-tailed paired t-tests. **G** t(8) = 0.1349, p = 0.8960; **H** t(8) = 0.1971, p = 0.8482; **I** t(8) = 0.1370, p = 0.8944; **J** t(8) = 0.2076, p = 0.8402; **K** t(8) = 1.827, p = 0.1051; **L** t(8) = 0.4631, p = 0.6556. **M** Mice were habituated for 5 min in the open field apparatus. Mice were then injected s.c. with 2 g/kg L-lactate or PBS before being placed back in the open field and movement was recorded for an additional 20 min. Locomotion is depicted in 1-minute bins. Data are presented as mean ± SEM. ns: non-significant. n = 9 mice per group. Significance was determined using Two-way ANOVA RM. L-lactate x Time: F(19, 160) = 1.250, p = 0.2247; L-lactate: F(19, 160) = 0.9351, p = 0.5409; Time: F(1, 160) = 39.64, p < 0.0001. Source data are provided as a Source Data file.

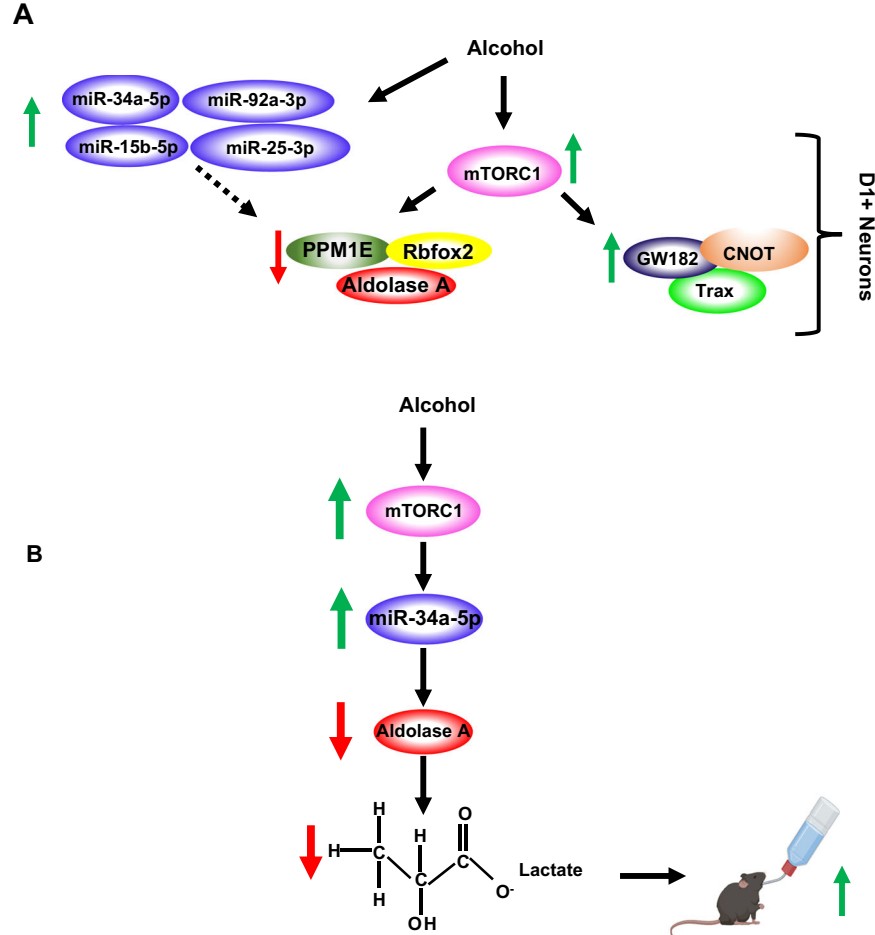

**Fig. 9 | Summary of results. A** Alcohol activates mTORC1 signaling in D1+ NAc neurons which in turn increases the translation of GW182, Trax and CNOT4 and represses the translation of Aldolase A, Rbfox2 and PPM1E. In parallel, alcohol increases the levels of miR-15b-5p, miR-25-3p, miR-92-3p and miR-34a-5p which are

predicted to target Aldolase A, Rbfox2 and PPM1E. **B** Alcohol activates mTORC1 signaling in the NAc which increases the level of miR-34a-5p, repressing the translation of Aldolase A and decreasing the level of L-lactate, promoting further drinking. Created in BioRender (2025) https://BioRender.com/88rbr00.

level of miR21, in turn promoting mTORC1-driven tumorigenesis[63]. Raab-Graham and colleagues previously reported that inhibition of rapamycin increased Kv1.1 synaptic levels in hippocampal dendrites[74] through miR129[75,76], although in contrast to our study, miR129 expression was mTORC1-independent[75,76].

We found that alcohol represses the translation of Aldolase A in the NAc. Aldolase A is an essential glycolytic enzyme that catalyzes the conversion of F1,6BP to G3P and DHAP[77]. Glucose metabolism is the major source of fuel in the brain[37]. Through multiple steps, glucose is metabolized to pyruvate which is converted to acetyl-CoA to produce ATP through the TCA cycle[37]. Our data suggest that alcohol reduces glycolysis and the TCA cycle in the NAc. These data are in

line with the findings by Volkow and colleagues that glucose metabolism in the brain is significantly decreased in subjects suffering from AUD[78–82]. In contrast to our results, mTORC1 was reported to increase glycolysis in skeletal muscle cells[83], macrophages[84], embryonic fibroblasts[85] and adipocytes[86]. However, since mTORC1 is generally activated during periods of nutrient availability, we cannot exclude a potential feedback loop mechanism to constrain cellular energetic needs through mTORC1-dependent miR upregulation and Aldolase A repression. Our data showing that glycolysis is inhibited by alcohol via mTORC1, provide yet another unexpected role for mTORC1 and suggest again that mTORC1 function depends on context and/or cell type. In line with this possibility, our data showing

that the activation of mTORC1 by alcohol in D1 NAc promotes the translation of microRNA machinery transcripts, resulting in the attenuation of Aldolase A transcription via miR-34a-5p which were also localized to D1 NAc neurons. Importantly, we found that miR34a-5p in D1 NAc neurons promotes alcohol but not sucrose intake. We cannot exclude the possibility that other targets of miR34a-5p in D1 NAc neurons contribute to alcohol drinking. However, putting together all our findings (Fig. 9), it is likely that the cell-specific attenuation of Aldolase A translation by miR34a-5p has a major contribution to neuroadaptations underlying alcohol drinking behavior. It will also be of interest to determine whether this cell-specific signaling pathway contributes to neuroadaptions that promote behaviors associated with other drugs of abuse.

Intriguingly, neuroadaptations that underlie addiction in general and AUD in specific[87] are high energy-consuming mechanisms that require ATP[88]. The finding that glycolysis is inhibited by alcohol bears the question of what is the alternative energy source that is required for alcohol-dependent neuroadaptations in the NAc. Interestingly, mTORC1 was reported to increase the pentose phosphate pathway which could be an alternative energy source for neurons[89]. Another possibility is that acetyl-CoA, instead of being generated by pyruvate, is produced by medium-chain fatty acids (MCFAs) through fatty acid beta-oxidation[90]. For instance, the medium-chain fatty acid octanoate can cross the blood-brain barrier[91], and has been shown to directly contribute ~20% of energy in the rat brain[92]. Thus, it is possible that when glucose levels are low, acetyl-CoA is produced through fatty acid metabolism. Another potential mechanism relates to acetate being metabolized from alcohol in the liver and astrocytes[93–95]. Acetate can be converted to acetyl-CoA which enters the TCA cycle. Thus, acetate may be an alternative energy source in the NAc.

In addition to pyruvate being converted to acetyl-CoA, pyruvate is also the source of lactate the end product of glycolysis[36,96]. We found that withdrawal from excessive alcohol intake reduces lactate content in the NAc, a process that requires mTORC1. This finding suggests a vectorial signaling in which alcohol withdrawal activates mTORC1 which increases the biogenesis of miR-34a-5p, in turn reducing the translation of Aldolase A leading to the attenuation of glycolysis and its final product, lactate (Fig. 9).

Lactate was initially thought to be produced solely by astrocytes and to be shuttled to neurons according to their energetic needs[97]. However, the lactate shuttle model has been disputed in part due to discrepancies in stoichiometry and kinetics of lactate production[98]. In fact, a large body of evidence have shown that glycolysis takes place in neurons[37,88,99–103], and we recently showed that hippocampal neurons metabolize glucose which is required for their normal function[102]. As mTORC1 activation by alcohol occurs essentially in D1 neurons and that rapamycin administration effectively reverses the alcohol-mediated reduction in lactate levels, our model strongly suggests a specific reduction of lactate levels within D1 neurons.

For many years lactate was thought to be only a byproduct of glycolysis[36]. However, recent studies showed that lactate is a signaling molecule and a substrate for epigenetic modification[36,101,104]. For example, secreted lactate is a ligand for the Gi-coupled hydroxycarboxylic acid receptor 1 (HCAR1)[105], and lactate binding to HCAR1 modulates neuronal network function and synaptic plasticity[105]. In addition, lactate was shown to be a substrate for lactylation, a newly identified posttranslational modification on proteins, including Histone H3 and Histone H1, resulting in enhanced gene transcription[106–108]. Thus, it is plausible that alcohol-mediated reduction of lactylation is the reason for the large number of transcripts that are reduced by alcohol in the NAc.

Reported that ketogenic diet which is given as a replacement for glucose reduces negative withdrawal symptoms in humans and mice[109–111] as well as alcohol intake in rats[110]. Interestingly, ketogenic diet was shown to decrease mTORC1 activity in the hippocampus and the liver[112,113]. These data, together with ours, suggest an intriguing possibility that, like the hypodopaminergic state in AUD subjects, which drives further drinking to alleviate allostasis symptoms, hypoglycolytic state also promotes further drinking to supply the brain with an alternative energy source. Further investigation on this topic is warranted.

Finally, we show that subcutaneous administration of L-lactate significantly reduces alcohol consumption. Attenuation of lactate levels or its shuttling between astrocytes and neurons is associated with stress[107] and depression[114,115]. Importantly, L-lactate administration reduces depressive-like symptoms in mice[116], and we found that systemic administration of L-lactate in mice reduces alcohol but not sucrose intake. Together, these data and ours suggest that psychiatric disorders are associated with an imbalance in lactate levels in the brain and that adjusting lactate levels reduces adverse effects associated with alcohol and depression. Furthermore, these data provide preclinical data to suggest that L-lactate could be developed as a new cost-effective readily available approach to treat AUD and potentially other psychiatric disorders.

In summary, this study unveils an unexpected dimension of mTORC1 function in alcohol-related behaviors. Specifically, our study unravels a paradoxical action of mTORC1 in orchestrating both translation repression and metabolic shift in response to chronic alcohol exposure which in turn drives further alcohol intake.

## Methods
### Ethics statement
All animal procedures were approved by UCSF Institutional Animal Care and Use Committee (IACUC) (animal protocol AN206967) and were conducted in agreement with the Association for Assessment and Accreditation of Laboratory Animal Care (AAALAC).

### Methods of euthanasia
Mice were euthanized by carbon dioxide inhalation followed by cervical dislocation, or deep anesthetization with pentobarbital (150 mg/kg) followed by transcranial perfusion with 4% paraformaldehyde. These procedures are in accordance with the Panel on Euthanasia of the American Veterinary Medical Association guidelines and with the standard operating procedures of the UCSF IACUC.

### Animals and breeding
Male C57BL/6 J mice were obtained from The Jackson Laboratory. Drd1a-Cre (D1-Cre) and AdoraA2-Cre (A2A-Cre) mice, both of which are on C57BL/6 background, were obtained from Mutant Mice Resource and Research Centers (MMRRC) UC Davis. Ribotag mice (ROSA26CAGGFP-L10a), which express the ribosomal subunit RPL10a fused to EGFP (EGFP-L10a) in Cre-expressing cells[117], were purchased from The Jackson Laboratory (B6;129S4-Gt (ROSA)26Sortm9(EGFP/Rpl10a)Amc/J). Ribotag mice were crossed with D1-Cre mice allowing EGFP-L10a expression in D1-expressing cells. Mouse genotype was determined by poly-chain reaction (PCR) analysis of tail DNA.

Only males (age 6-8 weeks old at the beginning of the experiments) were used in the study. Mice were individually housed on paper-chip bedding (Teklad #7084), under a reverse 12-hour light-dark cycle (lights on 1000 to 2200 h). Temperature and humidity were kept constant at 22 ± 2 °C, and relative humidity was maintained at 50 ± 5%. Mice were allowed access to food (Teklad Global Diet #2918) and tap water *ad libitum*.

### Plasmids generation and viral production
The mouse miR-34a nucleotide sequence (5'CCAGCTGTGAGTA ATTCTTTGGCAGTGT CTTAGCTGGTTGTTGTGAGTATTAGCTAAG GAAGCAATCAGCAAGTATACTGCCCTAGAAGTGCTGCACATTGT3') was synthesized. Synthesized DNA oligos containing the miRNA

sequences were annealed and inserted into pLL3.7 vector (11795, Addgene) at HpaI and XhoI sites. Plasmids were prepared using a Plasmid Maxi Kit. All constructs were verified by sequencing.

The production of miR-34a5p was conducted as described in ref. 19. Briefly, HEK293 lentiX cells (Clontech, Mountain View, CA) were transfected with the lentiviral packaging vectors psPAX2 and pMD2.G, together with the pLL3.7 miR-34a-5p or pLL3.7 GFP using lipofectamine 2000 in Opti-MEM medium. Six h after transfection, medium was replaced to DMEM-FBS 10%. Sixty h after transfection, supernatant containing the viral particles was collected, filtered into 0.22 μm filters and purified by ultracentrifugation at 26,000 g for 90 min at 4 °C. The pellet fraction containing the virus was resuspended in sterile PBS, aliquoted and stored at −80 °C until use. Virus titer was determined using the HIV-1 p24 antigen ELISA kit.

The plasmid and Adeno-associated virus (AAV)2-CMV-Flex-miR-34a-EGFP-WPRE (AAV2-Flex-miR-34a, titer: $2.89 \times 10^{13}$ GC/ml) were designed and produced by the VectorBuilder.

### Tissue harvesting
Mice were euthanized and brains were rapidly removed on an anodized aluminum block on ice. The NAc was isolated from a 1 mm thick coronal section located between +1.7 mm and +0.7 mm anterior to bregma according to the Franklin and Paxinos stereotaxic atlas (3rd edition). Collected tissues were immediately homogenized in 300 μl radioimmuno-precipitation assay (RIPA) buffer (50 mM Tris-HCl, pH 7.6, 150 mM NaCl, 2 mM EDTA, 1% NP-40, 0.1% SDS. and 0.5% sodium deoxycholate and protease and phosphatase inhibitors cocktail). Samples were homogenized by a sonic dismembrator. Protein content was determined using a BCA kit.

### Polysomal fractionation
Polysome-bound RNA was purified from mouse NAc according to a protocol we described previously[18]. Specifically, fresh mouse NAc was snap-frozen in a 1.5 ml Eppendorf tube and pulverized in liquid nitrogen with a pestle. After keeping on dry ice for 5 min, the powder of one NAc was resuspended in 1 ml lysis buffer (10 mM Tris pH 8.0, 150 mM NaCl, 5 mM MgCl$_2$, 1% NP40, 0.5% sodium deoxycholate, 40 mM dithiothreitol, 400U/ml Rnasin, 10 mM Ribonucleoside Vanadyl Complex and 200 μg/ml cycloheximide) followed by pipetting 20 times to further disrupt cell membranes. The homogenate was centrifuged for 10 seconds at $12,000 \times g$ to remove intact nuclei. The supernatant was collected, and ribosomes were further released by adding 2X extraction buffer (200 mM Tris pH7.5, 300 mM NaCl and 200 μg/ml cycloheximide). Samples were kept on ice for 5 min and then centrifuged at $12,000 \times g$, 4 °C for 5 min to remove mitochondria and membranous debris. The resulting supernatant was loaded onto a 15–45% sucrose gradient and centrifuged in a SW41Ti rotor at $250,000 \times g$, 4 °C for 2 h. Sucrose gradient fractions were collected and further digested with proteinase K solution (400 μg/ml proteinase K, 10 mM EDTA, 1% SDS) at 37 °C for 30 min, followed by phenol-chloroform extraction. RNA in the water phase of the polysomal fraction was recovered by ethyl alcohol precipitation. The integrity of the polysomal fractions is described in ref. 18. Specifically, RNA was visualized by migrating on a 1.5% agarose gel. Fractions 01–05 mainly contain tRNAs. Fractions 06–15 are enriched with 40S ribosomal subunit as well as 18S rRNA. Fractions 16–25 are enriched with 60S ribosomal subunit and 28S rRNA. Fractions 26–40 contain polysomal RNA[18,20].

### cDNA synthesis and real-time quantitative PCR (RT-qPCR)
Total RNA extracted from tissues was treated with DNase I. Synthesis of cDNA was performed using the iScript cDNA Synthesis Kit according to the manufacturer's instructions (Biorad). For polysomal RNA, and TRAP D1 RNA, cDNA synthesis and amplification were conducted using

Ovation RNA Amplification Kit V2 (Tecan). The resulting cDNA was used for quantitative RT-qPCR, using SYBR Green PCR Master mix (Thermo Fisher Scientific). Thermal cycling was performed on QuantStudio 5 real-time PCR System using a relative calibration curve. Each data point represents an average of 3 technical replicates. The quantity of each mRNA transcript was measured and expressed relative to Glyceraldehyde-3-Phosphate dehydrogenase (GAPDH). The Primers are listed in Supplementary Table 4.

### In Silico miR prediction
miRNA predictions were based on TargetScan 2[118], miRwalk[119] and miRDB[120,121]. miR-34a-5p and Aldolase A complimentary binding site was predicted using miRWalk[119]. miRWalk uses TarPmiR, a predictive algorithm, to generate a binding score[119]. Specifically, TarPmiR applies the trained random forest-based predictor to determine the target sites[122].

### miRNA extraction, cDNA synthesis and quantitative RT-qPCR
microRNAs were extracted from tissues using miRNeasy Mini Kit according to the manufacturer's instructions (Qiagen). miRNAs yield and purity was evaluated using a nanodrop ND-1000 spectrophotometer. cDNA synthesis was performed using the miRCURY LNA RT kit according to the manufacturer's instructions (Qiagen), starting with 500 ng of RNA and 5X reaction mix. Enzyme and nuclease-free water were added to a final volume of 20 μl. RNA spike-in of Unisp6 was added to each sample to monitor the efficiency of the reverse transcription reaction. Quantitative RT-qPCR was performed using miRCURY LNA SYBR Green master mix according to the manufacturer's instructions (Qiagen), and PCR samples were run on the QuantStudio 5. Each data point represents an average of 3 technical replicates. Relative microRNA expression was determined by the following calculation according to[123]. First, the average of cycle threshold (CT) of the miRNA of interest was subtracted from the CT of the control U6 rRNA to generate ΔCT. ΔCTs were then subtracted from the average of control ΔCT values for each sample to generate the ΔΔCT. miR levels were then calculated as 100 multiplied by $2^{-\Delta\Delta CT} \pm$ S.E.M. The Primers are listed in Supplementary Table 4.

### TRAP purification
After NAc dissection, D1 **or D2**-specific mRNAs were purified according to the established TRAP protocol[124]. The NAc was homogenized with a glass homogenizer in an ice-cold lysis buffer (150 mM KCl, 20 mM HEPES [pH 7.4], 10 mM MgCl2, 0.5 mM dithiothreitol, 100 μg/mL cycloheximide, 80U/μl RNasin Plus Rnase Inhibitor, and EDTA-free protease inhibitors). Following homogenization, samples were centrifuged at $2000 \times g$ at 4 °C for 10 min and the supernatant was removed to a new tube. NP-40 (final concentration 1%) and 1,2-Diheptanoylsn-glycero-3-phosphocholine (DHPC, final concentration 15 mM) were subsequently added and samples were incubated on ice for 5 min. Samples were centrifuged at 20,000 g at 4 °C for 10 min and the supernatant was transferred to a new tube. Streptavidin Dynabeads coated with biotin-linked mouse anti-GFP antibodies were then added to the supernatant and the samples were incubated overnight at 4 °C with end-over-end rotation. Beads were collected on a magnetic rack and washed three times with wash buffer (350 mM KCl, 20 mM HEPES pH 7.4, 10 mM MgCl$_2$, 0.5 mM dithiothreitol, 100 μg/mL cycloheximide, 1% NP-40). RNA was subsequently purified using the Absolutely RNA Isolation Nanoprep kit (Agilent). To ensure accurate quantitation, purified RNA was run on a Qubit 4 (Thermofisher). The validation of the RiboTag specificity is shown in Supplementary Fig. 13.

### Western blot analysis
Equal amounts of homogenates from individual mice (30 μg) were resolved on NuPAGE Bis-Tris gels (4–12% gradient, Life Technologies) and transferred onto nitrocellulose membranes (Millipore). Blots were blocked in 5% milk-PBS and 0.1% Tween 20 for 30 min

and then incubated overnight at 4 °C with primary antibodies. Membranes were then washed and incubated with HRP-conjugated secondary antibodies for 2 h at room temperature. Bands were visualized using Enhanced Chemiluminescence (ECL, Millipore). The optical density of the relevant band was quantified using ImageJ 1.44c software (NIH). Antibodies details are listed in Supplementary Table 5.

## Immunochemistry
Mice were deeply anesthetized with Euthasol and perfused with 0.9% NaCl, followed by 4% paraformaldehyde in PBS, pH 7.4. Brains were removed, post-fixed in the same fixative for 2 h, and transferred to PBS at 4 °C. On the following day, brains were transferred into 30% sucrose and stored for 3 days at 4 °C. Thirty μm-thick coronal sections were cut on a cryostat, collected serially and stored at -80 °C. Sections were permeabilized with, and blocked in, PBS containing 0.3% Triton and 5% donkey serum for 4 h. Sections were then incubated for 18 h at 4 °C on an orbital shaker with anti-pS6 (1:500) and anti-NeuN antibodies (1:500) diluted in 3% bovine serum albumin (BSA) in PBS. Next, sections were washed in PBS then incubated for 4 h with Alexa Fluor 596-labeled donkey anti-rabbit and Alexa Fluor 647-labeled donkey anti-mouse diluted in 3% BSA in PBS. After staining, sections were rinsed in PBS and cover slipped using Prolong Gold mounting medium (Thermofisher). Images were acquired using an Olympus Fluoview 3000 Confocal microscope using manufacture recommended filter configurations. Quantification was performed using the cell counter plugin in ImageJ software (NIH). Antibodies details are listed in Supplementary Table 5.

## Luciferase assay
The 3′ untranslated region (UTR) of Aldolase A containing miR-34a-5p predicted target site was cloned into the pmirGLO Dual-Luciferase miRNA Target Expression Vector (E1330, Promega). Mutant construct was generated with a mutated target site. HEK293 cells (ATCC) were seeded in 96-well plates and co-transfected with luciferase reporters, mimic miR-34a-5p (4464066, Invitrogen), or a mimic miR negative control (4464058, Invitrogen). Firefly (FL) and Renilla (RL) luciferase activities were measured 48 h after transfection using TECAN plate reader. Signal was calculated using FL/RL ratio relative to the empty pmirGLO reporter vector.

## NAc shell viral infection
Intra-NAc shell infusion of lentivirus or AVV2 was conducted as described in ref. 19. Briefly, mice were anesthetized using isoflurane. To avoid discomfort, distress, pain, and injury, mice received pre- and post-surgical monitoring and analgesics in accordance with the animal protocol. Stereotaxic surgeries will be completed within a sterile field to prevent infection and allowed to recover before experimentation. Bilateral viral infusions were done using stainless steel injectors (33 gauge, Hamilton) into the NAc shell (anteroposterior +1.2 mm, mediolateral ± 0.75 mm and dorsoventral -4.30 mm, from bregma). Animals were infused with either Ltv-Control expressing GFP only or Ltv-miR-34a-5p (1.2 × 10$^8$ pg/ml, 1 μl/side), or AAV2-Flex-control or AAV2-Flex-miR-34a (2.89 × 10$^{13}$ GC/ml, 1 μl/side) at an infusion rate of 0.1 μl/minute. After each infusion, the injectors were left in place for an additional 10 min to allow the virus to diffuse. Mice were carefully monitored for adverse effects due to experimental treatment.

## Preparation of solutions
Alcohol solution was prepared from absolute anhydrous alcohol (190 proof) diluted to 20% alcohol (v/v) in tap water. Sucrose solution (1%) was dissolved in tap water (w/v).

Rapamycin (20 mg/kg, R-5000, LC Laboratories) was dissolved in 5% DMSO and 95% saline. Vehicle contained 5% DMSO and 95% saline. Sodium L-Lactate (2 g/kg, Sigma Aldrich, 867-56-1) and NaCl (1 g/kg) were dissolved in PBS as described in ref. 41.

## Drug administration
Rapamycin: rapamycin (20 mg/kg) or vehicle was administered i.p. 3 h before the end of the last alcohol withdrawal session and tissues were harvested at the end of the 24 hour withdrawal session as in ref. 19.

L-lactate: L-lactate (2 g/kg) or NaCl (1 g/kg) was administered subcutaneously (s.c.). A "within-subject" design in which mice received both treatments in counterbalanced order, with one week in between treatments. Specifically, on weeks 8 and 9, mice were administered s.c. with L-lactate (2 g/kg), or vehicle solution 30 min before the beginning of the drinking session. On weeks 10 and 11, mice were systemically administered with NaCl (1 g/kg), or vehicle 30 min before the beginning of the drinking session. Alcohol and water consumption was evaluated at the end of 4 h and 24 h drinking session.

## Lactate measurement
The NAc lysates were added to 96-well plates and adjusted to 50 μl of reaction mix as described in the lactate colorimetric assay kit instructions (L-Lactate Assay Kit ab65331, Abcam). After incubation for 30 min at room temperature in the dark, the absorbance at 570 nm was measured using a microplate reader.

## Metabolomics
Metabolomics analysis was conducted as described in ref. 102. Specifically, fresh mouse NAc (water control samples: $n = 6$, alcohol withdrawal samples: $n = 7$) was snap-frozen in a 1.5 ml Eppendorf tube in liquid nitrogen. To extract the metabolites from the frozen tissue, samples were first homogenized in a cryogenic mortar and pestle before being mixed with 1 ml of 80% methanol chilled to −80 °C. Samples were then vortexed for 20 seconds and incubated at −80 °C for 20 min. Following the incubation, samples were vortexed for an additional 20 seconds and then centrifuged at 16,000 × $g$ for 15 min at 4 °C. The supernatant was transferred to a −80 °C prechilled tube. BCA assay was used to normalize the extracted metabolites to protein content. A 100 μg protein equivalent of extracted metabolites was aliquoted from each sample and dried in a CentriVap. The dried samples were then stored at −80 °C until analysis by the UCLA Metabolomics Center. Specifically, dried metabolites were resuspended in 50% ACN containing 2 uM internal standards (MSK-A2-1.2, Cambridge Isotope Laboratory) and 5 μl was loaded onto a Luna 3um NH2 100 A (150 × 2.0 mm) column (Phenomenex). The chromatographic separation was performed on a Vanquish Flex (Thermo Scientific) with mobile phases A (5 mM NH4AcO pH 9.9) and B (ACN) and a flow rate of 200 μl/min. The mobile phase composition changed linearly from 85% B to 5% B between 0-17 min, remained at 5% B for 9 min, returned in 1 min back to 85% B, and remained at 85% B for 11 min (column temperature: 27 °C). Metabolite detection was achieved with a Q Exactive mass spectrometer (Thermo Scientific) run with polarity switching (+3.5 kV/ − 3.4 kV) in full MS scan mode. RAW files were converted to mzXML format using the msConverter (ProteoWizard) and metabolite intensities was extracted with Maven (v 8.1.27.11).

## Drinking paradigm
**Two bottle choice - 20% alcohol.** Mice underwent 7 weeks of intermittent access to 20% (v/v) alcohol in a two-bottle choice drinking paradigm (IA20%2BC) as previously described[22]. Specifically, mice had 24-hour access to one bottle of 20% alcohol and one bottle of water on Mondays, Wednesdays, and Fridays, with alcohol drinking sessions starting 2 h into the dark cycle. During the 24 or 48 h (weekend) of alcohol withdrawal periods, mice had access to a bottle of water. The placement (right or left) of the bottles was alternated in each session to

control for side preference. Two bottles containing water and alcohol in an empty cage were used to evaluate the spillage. Alcohol and water intake were measured at the end of each 24 h drinking session (Supplementary Table 1).

**Two bottle choice - 1% sucrose.** Mice underwent 2 weeks of intermittent access to 1% (v/v) sucrose in a two-bottle choice drinking paradigm as previously described[19]. Specifically, mice had 24-hour access to one bottle of 1% sucrose and one bottle of water on Mondays, Wednesdays, and Fridays, with sucrose drinking sessions starting 2 h into the dark cycle. The placement (right or left) of the bottles was alternated in each session to control for side preference. Two bottles containing water and sucrose in an empty cage were used to evaluate the spillage. Sucrose and water intake were measured at the end of each 24-hour drinking session.

### Open field locomotion
Locomotion test was performed as described in ref. 125. Specifically, mice were habituated for 5 min in the open field apparatus (43 × 43 cm). Mice were then injected s.c. with 2 g/kg L-lactate or saline before being placed back in the open field apparatus and their movement was recorded for an additional 20 min. The test was performed using a "within-subject" design in which mice received both treatments in counterbalanced order. Mice were automatically video tracked using Ethovision XT software version 17 and locomotion per 1-minute bins was calculated.

### Glucose tolerance test
Glucose tolerance assay was performed as described previously[22]. Briefly, mice were deprived of food for 6 h and were then injected i.p. with 1 g/kg glucose. Blood samples were taken from a tail vein nick at different time intervals (0, 15, 30, 60, and 120 min post glucose administration), and blood glucose level was analyzed using a Bayer Contour blood glucose meter and test strips.

### Statistics & reproducibility
GraphPad Prism 7.0 (GraphPad Software Inc., La Jolla, CA) was used to plot and analyze the data. D'Agostino–Pearson normality and F-test/Levene tests were used to verify the normal distribution of variables and the homogeneity of variance, respectively. Data were analyzed using the appropriate statistical test, including two-tailed paired t-test, two-tailed unpaired t-test, one-way analysis of variance (ANOVA), and two-way ANOVA. ANOVAs were followed by post hoc tests as detailed in figure legends and Supplementary Table 2. Specifically, Sidak's post hoc test was used when performing specific planned comparisons rather than all pairwise comparisons, and Tukey's post hoc test was preferred when all possible pairwise comparisons were performed. All data are expressed as mean ± SEM or ± SD, and statistical significance was set at $p < 0.05$.

For every experiment, mouse groups were randomized. No statistical method was used to predetermine the sample size. Group size is provided in figure legends and in Supplementary Table 2. No data were excluded from the analyses. Two data points in drinking experiments are missing due to bottle spillage. Data collection was blinded when possible. Data analysis was blinded. For behavioral experiments, the investigators were blinded to the intervention methods for the mice.

### Reporting summary
Further information on research design is available in the Nature Portfolio Reporting Summary linked to this article.

## Data availability
The authors declare that all relevant data supporting the findings of this study are included in this published article, supplementary information files and source data file. The metabolomic data generated in this study have been deposited in the Zenodo database under accession code https://doi.org/10.5281/zenodo.15178256. Source data are provided in this paper. Source data are provided with this paper.

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

## Acknowledgements

This study was supported by the National Institute of Alcohol Abuse and Alcoholism, R01 AA027474 (D.R.), RF1 AG064170 and R01 AG065428 (K.N.) and F32 AG082460 (Y.J.S.). We thank Chhavi Shukla (University of California San Francisco) for technical assistance and Dr. Ellanor Whiteley for her contribution during the early stages of this study.

## Author contributions

S.L. Y.E., and D.R. conceived the work. Y.E. designed the study, conducted the in vivo experiments, and analyzed the data. K.P., A.S. and S.L. conducted the biochemical experiments and analyzed the data. D.S., S.G. and Z.W.H. participated in the behavioral experiments. Y.J.S. and K.N. designed, conducted the metabolomic experiment and analyzed the data. D.R. oversaw the study and wrote the manuscript together with Y.E.

## Competing interests

The authors declare no competing interests.
