## [Transparent Peer Review file · Nature Communications]

Paradoxical mTORC1-Dependent microRNA-mediated Translation Repression in the Nucleus Accumbens of Mice Consuming Alcohol Attenuates Glycolysis

Corresponding Author: Professor Dorit Ron

Version 0:

Reviewer comments:

Reviewer #1

(Remarks to the Author)

Ron and colleagues in their manuscript "Paradoxical mTORC1-Dependent microRNA-mediated Translation Repression in the Nucleus Accumbens of Mice Consuming Alcohol Attenuates Glycolysis" report a role for mTORC1 regulating glycolysis through the alcohol/mTORC1 dependent upregulation proteins that make up the microRNA machinery, an increase in microRNA expression and the repression of mRNA translation of specific mRNA that affect glycolysis. The experiments were carefully controlled and the evidence is overwhelmingly convincing. The manuscript will appeal to a broad audience, providing the first evidence of mechanistic understanding of how mTORC1 regulates glycolysis in a brain region specific manner. In addition, these studies provide a simple and inexpensive potential therapy for alcohol use disorder, a disease that affects ~11% of the US population (NIAAA).

While there is a tremendous amount of data included in the manuscript, there are a few concerns that if addressed will strengthen the manuscript.

Major Concerns:

1. Overexpression studies of miR-34a-5p are intriguing, and support a role for increasing alcohol intake. Does knockdown of miR-34a-5p rescue alcohol intake?
2. The D1 story is interesting; however, it is not followed up. A miR-34a-5p knockdown specifically in D1 neurons would tie the story together. Demonstrating that P-S6 is not upregulated in D2 labeled neurons provides an additional control. Alternatively, removing D⁺ albeit elegant may make the story more cohesive.

Minor Concerns:

1. The introduction reads more like a review of mTOR over providing an overview of why mTORC1 regulation of glycolysis is important, especially in the context of AUD. Since Nature Communications has a broad readership, background on microRNA machinery and microRNA function in the introduction is necessary to understand the context of the paper. Nowhere in the manuscript is there an explanation for the focus on NAc (center of motivated behavior).
2. mTOR's role in repression of protein expression is understudied. Few labs demonstrate a complete multidisciplinary story, making this story complete and novel. Additional citations to consider in the introduction and discussion are: Niere et al., MCP, 2016; Niere et al., PNAS, 2023; Ogorek et al., Hum Mol Genet, 2018, Liu et al., Oncogene, 2019.
4. The interpretation of the mRNA levels in Fig. S1 and S3 may be overstated. Total mRNA levels are a combination of mRNA stability and transcription. mTORC1 stabilizes mRNA (Sosanya et al., JBC, 2014). This fact does not change the importance of utilizing the translating polysome fraction for these studies. The wording needs to be changed.
3. There are missing citations in the discussion regarding NMDAR antagonism increasing mTOR activity (Li et al., Science, 2011; Workman et al., Neuropharmacology, 2013) and alcohol blocking NMDARs (Lovinger et al., 1989; Wolfe et al., 2016, etc.).

Reviewer #2

(Remarks to the Author)

The authors hypothesized that alcohol, by activating mTORC1, drives excessive alcohol intake in an animal model. They show that alcohol activated mTORC1 in the NAc, which in turn increased the translation of microRNA machinery transcripts and levels of miRs, including miR34a-5p. The alcohol-dependent activation of mTORC1 also repressed the translation of a

number of transcripts, including Aldolase A. The authors also showed that miR34a5p targets Aldolase A for translation repression. As a result of the alcohol-mediated mTORC1-dependent reduction of Aldolase A levels in the NAc, glycolysis was inhibited, and lactate levels were reduced. Thus, the hypothesis that one consequence of the mTORC1[up]-/miR34a-5p-[up]/Aldolase A[down]/glycolysis[down]/Lactate[down] pathway is the escalation of alcohol intake is supported and may lead to therapeutic advances in the future. The is a well-designed and -executed study, and the results are clear. The paper is well written and well discussed and makes a major contribution to the field. However, there is one issue that the authors need to consider or expand upon.

1. All of the work presented was in male subjects, and we have an obligation to examine mechanisms in both sexes. The authors could discuss this limitation further. They do state, "Our studies focused on male mice as mTORC1 is not activated by alcohol in the NAc of female mice," but it would be nice if this was explored further. Also, it would be interesting to see if one of the key observations in the mTORC1[up]-/miR34a-5p-[up]/Aldolase A[down]/glycolysis[down]/Lactate[down] pathway changes caused by alcohol, other than an increase in mTORC1 in the NAc, could be observed. For example, does lactate decrease alcohol consumption in female rodents?

Reviewer #3

(Remarks to the Author)

Paradoxical mTORC1-Dependent microRNA-mediated Translation Repression in the Nucleus Accumbens of Mice Consuming Alcohol Attenuates Glycolysis

NCOMMS-24-03504-T

Ehinger et al

This is a very interesting paper showing that alcohol consumption impacts mTORC1 activity in the nucleus accumbens (NAc) of mice, which modifies the translation of microRNA (miR)-relevant proteins. In turn, ethanol/mTORC1-regulated miRs control the expression of Aldolase, which regulates lactate levels in the NAc. It is proposed that lactate then regulates alcohol consumption. The investigator have employed an impressively multidisciplinary approach and the findings have considerable translational relevance. However, I have some comments/suggestions for the authors' considerable:

1. Introduction: Consider changing "environmental cues" to energy or metabolism-relevant stimuli or something similar.
2. Methods: It is unclear how polysomes were purified and polysome-containing lysates validated. Were ribosomal proteins enriched in the polysome fraction?
3. Methods: It is important to validate the D1-Cre x RiboTag mice (confirming that Cre expression was not 'leaky'). Was GFP in these animals expressed exclusively in D1 medium spiny neurons (as measured by RNAscope)? Were D2 transcripts harvested from these mice when GFP-RPL10 was immunoprecipitated?
4. Results: GAPDH was used to normalize mRNAs isolated from the polysome fraction. Thus, it is important to confirm that GAPDH expression is not altered by alcohol consumption.
5. Results: In figures 2 and 3, Western blots were cropped to show only putative bands representing Trax and Gw182. It is important to see the full gels/band distributions, along with molecular weight markers, to increase confidence that the bands indeed represent the proteins of interest.
6. Results: Some of the Western blots appear to have been cropped to eliminate bands (see Tubulin in Fig. S5).
7. Results: In figure 3, rather than an assessment of a small number of candidate gene transcripts associated with ribosomes in D1-Cre x RiboTag mice, it would be helpful to see RNAseq data from these animals (i.e., a broad assessment of the mRNAs undergoing translation in D1 MSNs in alcohol-exposed animals). Does rapamycin block the association of some/all of these ethanol-regulated mRNAs with the ribosomal complex?
8. Results: Were miR levels in Fig. 4B normalized to a housekeeping gene for each animal (for example, U6) prior to being expressed as % of water control? If so, it is important to confirm that U6 (or the relevant housekeeping gene) was not altered by alcohol. If not, how were absolute miR levels relative to RNA yield normalized between animals before being expressed as % of water control?
9. Results: Are miR34a-5p, miR15b-5p, miR25-3p, and/or miR92a-3p expressed in D1 MSNs?
10. Results: Why was miR127- 3p used as a control
11. Results: Is there a generalized increase in miR expression in the NAc of alcohol-exposed animals?
12. Results: It is unclear why mTORC1 activation would decrease in glycolysis. mTORC1 is generally activated during periods of nutrient availability. Does mTORC1 inhibition of Aldolase (via miR-mediated suppression) reflect a feedback loop to constrain cellular energetics?

13. Results: Ltv-miR34a appears to be non-cell type-specific; ideally, miR34a manipulations should be restricted to D1 MSNs.

14. Results: Does miR34a overexpression in the NAc decrease lactate levels?

15. Results: What is the effect of lactate administration on mTORC1 activity and miR34a expression in the NAc?

Reviewer #4

(Remarks to the Author)

The manuscript "Paradoxical mTORC1-Dependent microRNA-mediated Translation Repression in the Nucleus Accumbens of Mice Consuming Alcohol Attenuates Glycolysis" examines the role of mTOR-miRNA-Aldolase A in a mouse model of alcohol intake. Here the authors demonstrate that chronic alcohol leads to mTORC dependent adaptations including miR-34a. In particular, using a nice genetic model, the authors are able to show that these adaptations were largely occurring in D1-containing MSNs. Finally, the authors use a lenti-viral overexpression of miR-34a. enhances intake. Overall, this is a very carefully set of experiments. The data are well analyzed. However, there remains a general lack of connectiveness in the current manuscript. Perhaps one of the most important is that the authors make many conclusive statements about the role of mTORC1 which is largely through the use of Rapamycin. It is not in the least surprising that many proteins would be altered by the co-administration of rapamycin. This is even highlighted in the abstract where the authors themselves say "In parallel...." The connectivity between these processes is not convincing. Having said that, the data on miR-34a and downstream targets including metabolic processes is very nicely done. Some additional points to consider:

- 1) Are the D1-cre Ribotag used in 3C-D experiments the same lines as used in 3F. This was confusing. Please clarify.
- 2) The conclusions of the role of D1 are correlative. The authors clearly have the D1-cre mice and it is unclear why they did not perform the viral manipulations in a cell-type specific manner. This is a major weakness of the manuscript.
- 3) The overexpression data suggests that only one session is different. Is this the case? This is not very robust and may be due to lentiviral non-specificity.
- 4) Does overexpression or l-lactate alter any other aspect of alcohol induced behaviors?
- 5) The statistical table (S2) is very helpful and accurate.

Version 1:

Reviewer comments:

Reviewer #1

(Remarks to the Author)

The authors have addressed all concerns. The manuscript will be of interest to a broad audience.

Reviewer #3

(Remarks to the Author)

The authors have largely addressed the concerns that were raised about the original version of their manuscript. Overall, the findings are interesting and are likely to draw much interest in the field. Nevertheless, the major limitation remains the correlative nature of their findings. Specifically, direct cell type-specific manipulations of miR34a and related machineries (in D1 MSNs) were not performed. The including of new sequencing data from D2 MSNs helps to mitigate this concern. Nevertheless, considering the tools that are readily available for such cell type-specific manipulations, this remains an important gap in the paper.

Reviewer #4

(Remarks to the Author)

The authors have included a very detailed response to Reviewers 1-4 in the revised manuscript—which includes some additional data. The responses started off well. While the inclusion of a few words suggesting that there a direct link between mTORC and miR34a is requires further investigation in the discussion is appreciated, if this reviewer is still not convinced, but this is a matter of interpretation that should be left for the reader. However, the authors failed to address several comments and concerns raised by reviewer 4—that remain problematic for publication.

- 1) the findings for figure 7A are not convincing, and at best, may represent a change in tolerance. Further, in their response, the authors replied that the asterisk signifies the effect of the virus and no interaction of day. Please confirm that this was within subject analysis. To this end, the original comment suggested that the minimal effect was in part due to overexpression using lenti viral system that was not specific for neurons much less D1 or D2 neurons. This has not been addressed and weakens any conclusion that suggest as causative role for cell-type specific manner.
- 2) Given the limited and not robust data in figure 7, the question became are the additional data the authors have to

demonstrate the role of miR34a-5p in additional behaviors associated with alcohol use and abuse. Instead of including any measure that may have strengthened these arguments here, the authors included that they have recently obtained a grant. This does not help support the data in this manuscript for Nature Communications.

Version 2:

Reviewer comments:

Reviewer #4

(Remarks to the Author)

The authors have addressed the major concern in terms of cell type specificity. This was a very nice experiment and great addition to the manuscript.

However, the statistics question and presentation of statistics are not fixed or understood. How can F values be a decimal? What does mixed measures mean relative to repeated measures? How can the df of the f values be nearly identical in figure 7b and c given the number of measures and animals are more than 2x as many in figure 7c? Why are there no error bars for figure 7c.

What was the scientific rationale for using a Sidak's post hoc for some graphs and a Tukey's for other figures?

There is a general lack of completeness and carefulness in many of the reporting and presentation. While the information is interesting in this manuscript, and much work has gone into this, neither aspects are demonstrated by the way this manuscript or rebuttal(s) are constructed.

We thank the reviewers and the editor for the thoughtful and positive review of the manuscript. We conducted additional experiments and revised the manuscript carefully to address the questions and comments raised by the reviewers. We hope that our revised manuscript is ready for publication.

Below is a list of new experiments that we added to the revision:

- To address the D1-specificity question raised by Reviewers 1,2 and 4, we measured the level of the mTORC1-dependent identified transcripts in D2-specific polysomes. Specifically, we crossed A2A-Cre mice (expressing Cre in D2 neurons) with RiboTag mice, which enabled us to isolate mRNAs that are undergoing translation in D2 neurons. As shown in new **Supplementary Figure 5**, the translation of miR machinery transcripts Trax, GW182, and CNOT4 which were increased by alcohol in D1 neurons (**Figure 3**), were unaltered in D2 neurons. Similarly, the levels of Rbfox2, PPM1E, and Aldolase A whose translation was decreased by alcohol in D1 neurons (**Figure 3**), were not attenuated in D2 neurons (**Supplementary Figure 5**). Interestingly, Aldolase A translation was increased in D2 neurons. However, since global glycolysis is reduced by alcohol in the NAc of C57/Bl6 mice, we do not think that this change has any relevant functional consequences.
- To address Reviewer 2's question of whether glycolysis is altered in the NAc of female mice, we conducted an additional experiment and measured Aldolase A protein level in the NAc of females drinking alcohol or water. As shown in **Rebuttal Figure 1**, the protein level of Aldolase A is similar in the NAc of alcohol-drinking female mice vs. water controls. Aldolase A is an essential enzyme for glycolysis¹, and we and others found that mTORC1 is not activated in the NAc of female mice^{2,3}. Together, these data strongly suggest that alcohol does not alter glycolysis in the NAc of female mice. However, it is plausible that glycolysis is altered by alcohol in other brain regions of female mice. This possibility will be tested.
- To address Reviewer 3's question regarding the purity of the samples following TRAP ribosomal purification, we measured the level of *Drd1* and *Drd2* mRNA in samples from D1 and D2 neurons. As shown in **Suppl. Fig. 14**, there is a significant enrichment of *Drd1* mRNA in D1 neurons, whereas *Drd2* mRNA level is low. Conversely, we found a significant enrichment of *Drd2* mRNA in D2 neurons and a low level of *Drd1* mRNA. The low level of *Drd2* in D1 neurons and *Drd1* in D2 neurons is likely due to the D1/D2 co-expressing neurons found in the NAc⁴. These data indicate that the TRAP technique is reliable and there was no leakage of Cre expression.
- To address Reviewer 3's question on whether alcohol alters the translation of the housekeeping gene GAPDH, we measured its mRNA level in a polysomal fraction of D1 and D2 neurons, and the total protein level in the NAc of alcohol-drinking vs. water-drinking mice. We found that alcohol does not alter the translation of GAPDH (**Suppl. Fig. 1**), confirming our previous data showing that GAPDH levels are unaltered by alcohol in the brain (for example⁵⁻¹³). In addition, we also measured the levels of U6 in samples following miR purification using 5S as a control and found that U6 is also not altered by alcohol (**Suppl. Fig. 6**).

Reviewer #1 (Remarks to the Author):

Ron and colleagues in their manuscript “Paradoxical mTORC1-Dependent microRNA-mediated Translation Repression in the Nucleus Accumbens of Mice Consuming Alcohol Attenuates Glycolysis” report a role for mTORC1 regulating glycolysis through the alcohol/mTORC1 dependent upregulation proteins that make up the microRNA machinery, an increase in microRNA expression and the repression of mRNA translation of specific mRNA that affect glycolysis. The experiments were carefully controlled and the evidence is overwhelmingly convincing. The manuscript will appeal to a broad audience, providing the first evidence of a mechanistic understanding of how mTORC1 regulates glycolysis in a brain region-specific manner. In addition, these studies provide a simple and inexpensive potential therapy for alcohol use disorder, a disease that affects ~11% of the US population (NIAAA). While there is a tremendous amount of data included in the manuscript, there are a few concerns that, if addressed, will strengthen the manuscript.

Major Concerns:

1. Overexpression studies of miR-34a-5p are intriguing and support a role for increasing alcohol intake. Does knockdown of miR-34a-5p rescue alcohol intake?

Answer: The reviewer raises an interesting point. As far as we know, the shRNA strategy, which we have a lot of experience with, is not used to target miRs. CRISPR-Cas9 has been used to knockdown specific miRs. Unfortunately, we have not had success with CRISPR-Cas9 to knock down genes. Therefore, we cannot address this question at this point. In future experiments, we plan to use mice expressing Cas9 only in the presence of Cre which will enable us to answer this interesting question.

2. The D1 story is interesting; however, it is not followed up. A miR-34a-5p knockdown specifically in D1 neurons would tie the story together. Demonstrating that P-S6 is not upregulated in D2 labeled neurons provides an additional control. Alternatively, removing D1—albeit elegant—may make the story more cohesive.

Answer: As mentioned above, we cannot knock down miRs in a cell type-specific manner. Therefore, to address the cell-type specificity question, we conducted a new experiment in which we tested the levels of the miR machinery transcripts as well as the putative miR targets in the polysomal fraction of NAc D2 neurons of alcohol and water-drinking mice. We found that the miR machinery transcripts GW182, Trax and CNOT4 are unaltered by alcohol in D2 neurons (**Suppl. Fig. 5**). Furthermore, miRs targets, including Aldolase A targeted by miR34a5p are also unchanged by alcohol in D2 neurons (**Suppl. Fig. 5**). These data, together with our present (**Fig. 3C-D**) and previous data¹⁴ showing that mTORC1 activity is localized almost solely to D1 NAc MSNs strengthen our conclusion that mTORC1 \uparrow /miR34a5p \uparrow /Aldolase A \downarrow pathway is localized to D1 NAc neurons. Nevertheless, we added a caveat of the lack of cell-type specificity in our experiments testing miR34a function in the discussion.

Minor Concerns:

1. The introduction reads more like a review of mTOR over providing an overview of why mTORC1 regulation of glycolysis is important, especially in the context of AUD. Since Nature Communications has a broad readership, background on microRNA machinery and microRNA function in the introduction is necessary to understand the context of the paper. Nowhere in the manuscript is there an explanation for the focus on NAc (center of motivated behavior).

Answer: The reviewer is correct, and we apologize for the oversight. To address the comment, we expanded the background on microRNA biology in the introduction section. The focus on the NAc stems from our discovery that alcohol activates the H-Ras/PI3K/AKT pathway in the NAc of alcohol consuming rodents (refs). As this cascade is upstream of mTORC1, we measured

mTORC1 activity in this brain region and found that it is activated by alcohol binge and withdrawal specifically in the shell of the NAc¹⁰. Furthermore, we showed that alcohol-dependent activation mTORC1 in the NAc promotes structural and synaptic plasticity leading to NAc-dependent behaviors such as excessive alcohol intake, seeking and reward^{11,13,15}. Finally, we found that inhibition of mTORC1, specifically in the NAc, attenuates alcohol intake¹³. We added this paragraph to the introduction.

2. mTOR's role in repression of protein expression is understudied. Few labs demonstrate a complete multidisciplinary story, making this story complete and novel. Additional citations to consider in the introduction and discussion are: Niere et al., MCP, 2016; Niere et al., PNAS, 2023; Ogorek et al., Hum Mol Genet, 2018, Liu et al., Oncogene, 2019.

Answer: We thank the reviewer for the kind words and for pointing out these publications, we are sorry for the oversight. We updated the discussion, which now includes a description of these key findings.

3. The interpretation of the mRNA levels in Fig. S1 and S3 may be overstated. Total mRNA levels are a combination of mRNA stability and transcription. mTORC1 stabilizes mRNA (Sosanya et al., JBC, 2014). This fact does not change the importance of utilizing the translating polysome fraction for these studies. The wording needs to be changed.

Answer: We thank the reviewer for pointing this out, and we changed “transcription” to “transcription and/or RNA stability”.

4. There are missing citations in the discussion regarding NMDAR antagonism increasing mTOR activity (Li et al., Science, 2011; Workman et al., Neuropharmacology, 2013) and alcohol blocking NMDARs (Lovinger et al., 1989; Wolfe et al., 2016, etc.).

Answer: Li et al. was included in the original discussion. We apologize for omitting the other information which was added to the revision.

Reviewer #2 (Remarks to the Author):

The authors hypothesized that alcohol, by activating mTORC1, drives excessive alcohol intake in an animal model. They show that alcohol activated mTORC1 in the NAc, which in turn increased the translation of microRNA machinery transcripts and levels of miRs, including miR34a-5p. The alcohol-dependent activation of mTORC1 also repressed the translation of a number of transcripts, including Aldolase A. The authors also showed that miR34a5p targets Aldolase A for translation repression. As a result of the alcohol-mediated mTORC1-dependent reduction of Aldolase A levels in the NAc, glycolysis was inhibited, and lactate levels were reduced. Thus, the hypothesis that one consequence of the mTORC1^{up}/miR34a-5p^{up}/Aldolase A^{down}/glycolysis^{down}/Lactate^{down} pathway is the escalation of alcohol intake is supported and may lead to therapeutic advances in the future. The study is a well-designed and -executed study, and the results are clear. The paper is well written and well discussed and makes a major contribution to the field. However, there is one issue that the authors need to consider or expand upon.

1. All of the work presented was in male subjects, and we have an obligation to examine mechanisms in both sexes. The authors could discuss this limitation further. They do state, “Our studies focused on male mice as mTORC1 is not activated by alcohol in the NAc of female mice,” but it would be nice if this was explored further.

Answer: To address the reviewer’s comment, we expanded the discussion about the differences in male vs. female mice. We are currently studying the underlying mechanisms of sex differences in mTOR signaling.

2. it would be interesting to see if one of the key observations in the mTORC1[up][miR34a-5p[up]/Aldolase A[down]/glycolysis[down]/Lactate[down] pathway changes caused by alcohol, other than an increase in mTORC1 in the NAc, could be observed. For example, does lactate decrease alcohol consumption in female rodents?

Answer: Our reasoning for examining this pathway in male mice stemmed from our findings that mTORC1, as the reviewer pointed out, is not activated in adult female mice³. Similarly, mTORC1 is not activated in the OFC and mPFC of adult female mice³. Furthermore, inhibition of mTORC1 by systemic administration³ or intra NAc infusion², of the selective mTORC1 inhibitor, rapamycin, does not alter alcohol intake in females³. These data strongly suggest that the abovementioned mTORC1-dependent pathway in the NAc neurons does not produce neuronal adaptations detected in male mice. To confirm this conclusion, we conducted an additional experiment in which we measured the level of Aldolase A protein levels in alcohol drinking vs. water drinking female mice. In contrast to males, Aldolase A levels were unchanged by alcohol drinking in females (**Rebuttal Figure 1**). These data suggest that this signaling cascade is specifically relevant to males. However, it is plausible that mTORC1 is activated in females by alcohol in other regions and/or cell types. We are currently conducting a survey of alcohol-mediated mTORC1 activation patterns in female mice. Preliminary data suggest that alcohol may activate mTORC1 in a subset of microglial cells in the NAc of female mice and we are currently exploring this possibility. This is a large endeavor which is outside the scope of this manuscript.

Reviewer #3 (Remarks to the Author):

This is a very interesting paper showing that alcohol consumption impacts mTORC1 activity in the nucleus accumbens (NAc) of mice, which modifies the translation of microRNA (miR)-relevant proteins. In turn, ethanol/mTORC1-regulated miRs control the expression of Aldolase, which regulates lactate levels in the NAc. It is proposed that lactate then regulates alcohol consumption. The investigators have employed an impressively multidisciplinary approach, and the findings have considerable translational relevance. However, I have some comments/suggestions for the authors' considerable:

1. Introduction: Consider changing "environmental cues" to energy or metabolism-relevant stimuli or something similar.

Answer: We thank the reviewer for this comment and changed the text to metabolism-relevant stimuli.

2. Methods: It is unclear how polysomes were purified and polysome-containing lysates validated. Were ribosomal proteins enriched in the polysome fraction?

Answer: We expanded the description of the polysomal fractionation in the method as we previously described^{11,12,14}.

Rebuttal Figure 2 from our previous publication depicts the confirmation of the technique (Liu et al. Molecular Psychiatry

Supplemental Figure 1). Furthermore, herein we used two different methods (polysome fractionation and TRAP GFP pulldown) to measure transcripts undergoing translation. Using both methods, we found that microRNA machinery transcripts are increased, whereas other transcripts, including Aldolase A, are decreased in the NAc of alcohol-drinking mice in an mTORC1-dependent manner.

3. Methods: It is important to validate the D1-Cre x RiboTag mice (confirming that Cre expression was not 'leaky'). Was GFP in these animals expressed exclusively in D1 medium spiny neurons (as measured by RNAscope)? Were D2 transcripts harvested from these mice when GFP-RPL10 was immunoprecipitated?

Answer: The D1-Cre¹⁶⁻¹⁸ and the RiboTag mice¹⁹⁻²¹ have been widely used by the neuroscience community without showing any leak. Nevertheless, we performed a new experiment in which the levels of *Drd1* and *Drd2* were measured in the D1 and D2-specific mRNA fractions (**Suppl. Fig. 14**). The results show a clear and clean polysomal fractionation isolation. In addition, to address reviewer 1 point 2, we isolated D2R-cells transcript and found that the increase of translation of the 3 miR machinery proteins and the reduction in Aldolase A, PPM1E, and Rbfox2 do not occur in D2 neurons, demonstrating the D1-specificity of this pathway. Therefore, we are very confident in the mouse lines used in this study.

4. Results: GAPDH was used to normalize mRNAs isolated from the polysome fraction. Thus, it is important to confirm that GAPDH expression is not altered by alcohol consumption.

Answer: To address the reviewer's valid point, we normalized *GAPDH* to another housekeeping gene, *HPRT*. **Suppl. Fig. 1A-B** clearly shows that alcohol consumption does not change *GAPDH* levels in the D1 and D2 polysome fractions. We also measured *GAPDH* protein level and normalized it to Tubulin and found that alcohol consumption does not alter the protein level of *GAPDH* as well (**Suppl. Fig. 1C-D**). These data are in line with previous data from our lab that show that *GAPDH* protein level is not altered by alcohol in the brain (for example⁵⁻¹³).

5. Results: In figures 2 and 3, Western blots were cropped to show only putative bands representing Trax and Gw182. It is important to see the full gels/band distributions, along with molecular weight markers, to increase confidence that the bands indeed represent the proteins of interest.

Answer: The full membranes is now shown in **Rebuttal Fig. 3 and 4**.

Rebuttal Figure 3: Raw data associated with Figure 1E depicting alcohol-mediated increase in the level of Trax

Rebuttal figure 4: Raw data associated with Figure 1F depicting alcohol-mediated increase in the level of GW182 (Trnc6a)

Rebuttal figure 5: Raw data associated with Figure S5 depicting the absence of alteration by alcohol in levels of Aldolase C in the NAC

6. Results: Some of the Western blots appear to have been cropped to eliminate bands (see Tubulin in Fig. S5).

Answer: The full membrane is shown in Rebuttal Fig. 5, and we decided to only show water and withdrawal and not binge drinking. We apologize and we modified the figure to avoid any confusion.

7. Results: In figure 3, rather than an assessment of a small number of candidate gene transcripts associated with ribosomes in D1-Cre x RiboTag mice, it would be helpful to see RNAseq data from these animals (i.e., a broad assessment of the mRNAs undergoing translation in D1 MSNS in alcohol-exposed animals). Does rapamycin block the association of some/all of these ethanol-regulated mRNAs with the ribosomal complex?

Answer: a. Our strategy was to first identify transcripts whose translation is altered by alcohol in an mTORC1-dependent manner in the NAC using bulk polysomal-RNAseq. Surprisingly, the number of transcripts that were upregulated (12 transcripts)¹¹ or downregulated (32 transcripts) (data herein) was very limited. We then confirmed the D1-

specificity of selected candidates using the D1-Cre x Ribotag mice and the TRAP technique.

b. Rapamycin does not directly block the association of mRNA with the ribosomal complex. Rapamycin is a very selective allosteric mTORC1 inhibitor which binds to FKBP12. This rapamycin-FKBP12 complex then binds to a specific domain in the C-terminus of mTOR leading to mTORC1 inhibition²². This inhibition abolishes the activation of the downstream targets 4EBP and S6K. 4EBP and S6K are key regulators of the translation initiation, the major rate-limiting step in mRNA translation²².

8. Results: Were miR levels in Fig. 4B normalized to a housekeeping gene for each animal (for example, U6) prior to being expressed as % of water control? If so, it is important to confirm that U6 (or the relevant housekeeping gene) was not altered by alcohol. If not, how were absolute miR

levels relative to RNA yield normalized between animals before being expressed as % of water control?

Answer: miR levels in Fig. 4B were normalized to U6. We apologize for the oversight. We added U6 to Fig. 4B. In addition, to address the reviewer's point, we conducted an additional experiment in which we normalized U6 levels to 5S and found that U6 levels are not changed by alcohol (**Suppl. Fig. 6**).

9. Results: Are miR34a-5p, miR15b-5p, miR25-3p, and/or miR92a-3p expressed in D1 MSNs?

Answer: The TRAP method allows us to isolate actively translating mRNAs in a cell specific manner; it does not, however, enable the isolation miRs in a cell-type-specific manner. Furthermore, unfortunately, miR-RNAseq does not exist. However, since the translation of miR machinery proteins is not altered in D2 neurons, as shown by the new data (**Suppl. Fig. 5**), the upregulation of these miRs is likely to happen specifically in D1 neurons.

10. Results: Why was miR127- 3p used as a control

Answer: We selected miR127-3p as a control because it does not target any of the 32 downregulated transcripts.

11. Results: Is there a generalized increase in miR expression in the NAc of alcohol-exposed animals?

Answer: We thank the reviewer for this important question. Our data suggest that alcohol, via mTORC1, increases the expression of a subset of specific miRs. As shown in **New Figure 4**, in addition to miR127-3p, the levels of miR15a-5p, miR122-5p and miR19-3p were also unaltered by alcohol.

12. Results: It is unclear why mTORC1 activation would decrease in glycolysis. mTORC1 is generally activated during periods of nutrient availability. Does mTORC1 inhibition of Aldolase (via miR-mediated suppression) reflect a feedback loop to constrain cellular energetics?

Answer: This is a very good point which we believe is one of the reasons that make our findings so novel. We added this point to the discussion. We believe that mTORC1 inhibition or activation of glycolysis depends on the context, the environment and/or cell type.

13. Results: Ltv-miR34a appears to be non-cell type-specific; ideally, miR34a manipulations should be restricted to D1 MSNs.

Answer: In our new experiment (**Suppl. Fig. 5**), we found that alcohol-dependent translation of Rbfox2, PPM1E, and Aldolase A was unaltered in D2 MSNs, suggesting that the mTORC1/miR34a pathway is specifically activated in D1 MSNs. Therefore, we believe that the effect of the Ltv-miR34a on alcohol consumption is via miR34a overexpression in D1 MSNs.

14. Results: Does miR34a overexpression in the NAc decrease lactate levels?

Answer: We did not measure lactate level after miR34a overexpression. However, we are currently conducting a study focusing on the role of Aldolase A in the NAc. We overexpressed a Aldolase A, the downstream target of miR34a-5p, in the NAc of mice and measured lactate after 7 weeks of alcohol drinking. We found that Aldolase A overexpression rescues the lactate levels in alcohol drinking mice (**Rebuttal Figure 6**), and since miR34a-5p overexpression decreases Aldolase A protein levels in cells and *in vivo*, these data suggest that mTORC1 activation by alcohol decreases lactate levels in the NAc via the miR34a/Aldolase A pathway.

15. Results: What is the effect of lactate administration on mTORC1 activity and miR34a expression in the NAc?

Answer: This is an interesting point. Lactate administration replenishes the level of endogenous lactate (**Rebuttal Figure 7**), which is reduced by alcohol through the miR machinery. As lactate administration decreases alcohol intake in mice, we cannot determine if changes in mTORC1 activity or miR34a in response to lactate administration are due to lactate *per se* or because alcohol levels are lower in the brain of lactate-treated animals vs. control animals.

Reviewer #4 (Remarks to the Author):

The manuscript “Paradoxical mTORC1-Dependent microRNA-mediated Translation Repression in the Nucleus Accumbens of Mice Consuming Alcohol Attenuates Glycolysis” examines the role of mTOR-miRNA-Aldolase A in a mouse model of alcohol intake. Here the authors demonstrate that chronic alcohol leads to mTORC dependent adaptations including miR-34a. In particular, using a nice genetic model, the authors are able to show that these adaptations were largely occurring in D1-containing MSNs. Finally, the authors use a lenti-viral overexpression of miR-34a. enhances intake. Overall, this is a very carefully set of experiments. The data are well analyzed.

However, there remains a general lack of connectiveness in the current manuscript. Perhaps one of the most important is that the authors make many conclusive statements about the role of mTORC1 which is largely through the use of Rapamycin. It is not in the least surprising that many proteins would be altered by the co-administration of rapamycin. This is even highlighted in the abstract where the authors themselves say “In parallel...” The connectivity between these processes is not convincing. Having said that, the data on miR-34a and downstream targets including metabolic processes is very nicely done.

Answer: We respectfully disagree with the reviewer. Since rapamycin is a very selective and potent mTORC1 inhibitor²³, we as a tool. As shown in our previous study¹¹ and herein, the translation of only 44 transcripts was altered by alcohol in a mTORC1 dependent manner (12 are upregulated and 32 are downregulated). Furthermore, it is rather surprising that 32 transcripts were downregulated by alcohol in an mTORC1-dependent manner, and as reviewer 1 pointed out, the role of mTORC1 in protein translation repression is understudied, thereby making this study novel. Nevertheless, we changed the text and stated that the direct link between the upregulation of miR machinery transcripts by alcohol and repression of translation also by alcohol requires further investigation.

Some additional points to consider:

1) *Are the D1-cre Ribotag used in 3C-D experiments the same lines as used in 3F. This was confusing. Please clarify.*

Answer: The D1-Cre Ribotag mice are the same in **3C-D** and **3F**.

2) *The conclusions of the role of D1 are correlative. The authors clearly have the D1-cre mice and it is unclear why they did not perform the viral manipulations in a cell-type specific manner. This is a major weakness of the manuscript.*

Answer: To address this point, together with our finding that mTORC1 is almost only activated in D1 neurons, we isolated polysomal mRNAs from D2 neurons and showed that the increase of translation of the 3 miR machinery proteins and the decrease in translation of the 3 transcripts does not occur in D2 neurons suggesting a D1 specificity of this pathway (**Suppl. Fig. 5**).

3) *The overexpression data suggests that only one session is different. Is this the case? This is not very robust and may be due to lentiviral non-specificity.*

Answer: The statistical difference shown in **Figure 7** represents the main effect of the virus.

4) *Does overexpression or L-lactate alter any other aspect of alcohol induced behaviors?*

Answer: We have an NIH grant proposal focused on alcohol and glycolysis, which includes a detailed analysis of L-lactate and AUD-related phenotypes. These experiments will start once the grant is awarded.

References

- 1 Pirovich, D. B., Da'dara, A. A. & Skelly, P. J. Multifunctional Fructose 1,6-Bisphosphate Aldolase as a Therapeutic Target. *Front Mol Biosci* **8**, 719678, doi:10.3389/fmolb.2021.719678 (2021).
- 2 Cozzoli, D. K. *et al.* Functional regulation of PI3K-associated signaling in the accumbens by binge alcohol drinking in male but not female mice. *Neuropharmacology* **105**, 164-174, doi:10.1016/j.neuropharm.2016.01.010 (2016).
- 3 Ehinger, Y., Phamluong, K. & Ron, D. Sex differences in the interaction between alcohol and mTORC1. *bioRxiv*, 2023.2010.2004.560781, doi:10.1101/2023.10.04.560781 (2023).
- 4 Bertran-Gonzalez, J. *et al.* Opposing patterns of signaling activation in dopamine D1 and D2 receptor-expressing striatal neurons in response to cocaine and haloperidol. *J Neurosci* **28**, 5671-5685, doi:10.1523/JNEUROSCI.1039-08.2008 (2008).
- 5 Ben Hamida, S. *et al.* Protein tyrosine phosphatase alpha in the dorsomedial striatum promotes excessive ethanol-drinking behaviors. *J Neurosci* **33**, 14369-14378, doi:10.1523/JNEUROSCI.1954-13.2013 (2013).
- 6 Gibb, S. L., Hamida, S. B., Lanfranco, M. F. & Ron, D. Ethanol-induced increase in Fyn kinase activity in the dorsomedial striatum is associated with subcellular redistribution of protein tyrosine phosphatase alpha. *J Neurochem* **119**, 879-889, doi:10.1111/j.1471-4159.2011.07485.x (2011).
- 7 Wang, J. *et al.* Long-lasting adaptations of the NR2B-containing NMDA receptors in the dorsomedial striatum play a crucial role in alcohol consumption and relapse. *J Neurosci* **30**, 10187-10198, doi:10.1523/JNEUROSCI.2268-10.2010 (2010).
- 8 Ben Hamida, S. *et al.* Mammalian target of rapamycin complex 1 and its downstream effector collapsin response mediator protein-2 drive reinstatement of alcohol reward seeking. *Addict Biol* **24**, 908-920, doi:10.1111/adb.12653 (2019).

- 9 Barak, S. *et al.* Disruption of alcohol-related memories by mTORC1 inhibition prevents relapse. *Nat Neurosci* **16**, 1111-1117, doi:nn.3439 [pii] 10.1038/nn.3439 (2013).
- 10 Laguesse, S., Morisot, N., Phamluong, K. & Ron, D. Region specific activation of the AKT and mTORC1 pathway in response to excessive alcohol intake in rodents. *Addict Biol* **22**, 1856-1869, doi:10.1111/adb.12464 (2017).
- 11 Laguesse, S. *et al.* Prosapip1-Dependent Synaptic Adaptations in the Nucleus Accumbens Drive Alcohol Intake, Seeking, and Reward. *Neuron* **96**, 145-159 e148, doi:10.1016/j.neuron.2017.08.037 (2017).
- 12 Liu, F. *et al.* mTORC1-dependent translation of collapsin response mediator protein-2 drives neuroadaptations underlying excessive alcohol-drinking behaviors. *Mol Psychiatry* **22**, 89-101, doi:10.1038/mp.2016.12 (2017).
- 13 Neasta, J., Ben Hamida, S., Yowell, Q., Carnicella, S. & Ron, D. Role for mammalian target of rapamycin complex 1 signaling in neuroadaptations underlying alcohol-related disorders. *Proc Natl Acad Sci U S A* **107**, 20093-20098, doi:10.1073/pnas.1005554107 (2010).
- 14 Beckley, J. T. *et al.* The First Alcohol Drink Triggers mTORC1-Dependent Synaptic Plasticity in Nucleus Accumbens Dopamine D1 Receptor Neurons. *J Neurosci* **36**, 701-713, doi:10.1523/JNEUROSCI.2254-15.2016 (2016).
- 15 Ben Hamida, S. *et al.* The small G protein H-Ras in the mesolimbic system is a molecular gateway to alcohol-seeking and excessive drinking behaviors. *J Neurosci* **32**, 15849-15858, doi:10.1523/JNEUROSCI.2846-12.2012 (2012).
- 16 Wang, J. *et al.* Alcohol Elicits Functional and Structural Plasticity Selectively in Dopamine D1 Receptor-Expressing Neurons of the Dorsomedial Striatum. *J Neurosci* **35**, 11634-11643, doi:10.1523/JNEUROSCI.0003-15.2015 (2015).
- 17 Zhang, J. *et al.* c-Fos facilitates the acquisition and extinction of cocaine-induced persistent changes. *J Neurosci* **26**, 13287-13296, doi:10.1523/JNEUROSCI.3795-06.2006 (2006).
- 18 Wei, X. *et al.* Dopamine D1 or D2 receptor-expressing neurons in the central nervous system. *Addict Biol* **23**, 569-584, doi:10.1111/adb.12512 (2018).
- 19 Zhou, P. *et al.* Interrogating translational efficiency and lineage-specific transcriptomes using ribosome affinity purification. *Proc Natl Acad Sci U S A* **110**, 15395-15400, doi:10.1073/pnas.1304124110 (2013).
- 20 Dalal, J. S. *et al.* Loss of Tsc1 in cerebellar Purkinje cells induces transcriptional and translation changes in FMRP target transcripts. *Elife* **10**, doi:10.7554/eLife.67399 (2021).
- 21 Kilfeather, P. *et al.* Single-cell spatial transcriptomic and translational profiling of dopaminergic neurons in health, aging, and disease. *Cell Rep* **43**, 113784, doi:10.1016/j.celrep.2024.113784 (2024).
- 22 Hausch, F., Kozany, C., Theodoropoulou, M. & Fabian, A. K. FKBP5 and the Akt/mTOR pathway. *Cell Cycle* **12**, 2366-2370, doi:10.4161/cc.25508 (2013).
- 23 Li, J., Kim, S. G. & Blenis, J. Rapamycin: one drug, many effects. *Cell Metab* **19**, 373-379, doi:10.1016/j.cmet.2014.01.001 (2014).

We thank the editor and the reviewers for their thoughtful additional comments.

We revised the manuscript according to the reviewers' suggestions. Our responses to the comments are detailed below. The original comments of each reviewer are in *italics*. Additional text in the manuscript is highlighted in bold.

Listed below are the new figures from the experiments we conducted to address the reviewers' comments.

- Alcohol drinking is increased in D1-Cre mice overexpressed with miR-34a in the NAc **Figure 7B**
- miR-34a overexpression in NAc D1 neurons does not alter locomotion in **Figure 7C**
- Overexpression of miR34a in D1 NAc neurons does not alter sucrose intake **Figure 7D**
- AAV-Flex-miR-34a characterization **Supplementary Figure 11**

Reviewer #1 (Remarks to the Author):

The authors have addressed all concerns. The manuscript will be of interest to a broad audience.

Reviewer #3 (Remarks to the Author):

The authors have largely addressed the concerns that were raised about the original version of their manuscript. Overall, the findings are interesting and are likely to draw much interest in the field. Nevertheless, the major limitation remains the correlative nature of their findings. Specifically, direct cell type-specific manipulations of miR-34a and related machineries (in D1 MSNs) were not performed. The including of new sequencing data from D2 MSNs helps to mitigate this concern. Nevertheless, considering the tools that are readily available for such cell type-specific manipulations, this remains an important gap in the paper.

Answer: To address this comment, we performed a cell-specific manipulation of miR-34a levels followed by a behavioral experiment. Specifically, the NAc of D1-Cre male mice was bilaterally infused with an AAV-flex-miR-34a in order to specifically overexpress miR-34a5p in D1 neurons. As shown in **Figure 7B (Rebuttal Figure 1B)**, overexpressing of miR-34a-5p in D1 neurons accelerates the escalation of alcohol drinking. Interestingly, the effect was more pronounced with D1-specific manipulation compared to broad neuronal infection of miR34a (**Rebuttal Figure 1A**). We show that the increase in alcohol intake is not due to hyperlocomotion as overexpression of miR-34a-5p in NAc D1 neurons did not alter locomotion (**Figure 7C**) Furthermore, we show that the increase in intake is specific for alcohol as the intake of sucrose was unaltered in D1-Cre mice subjected to overexpression of miR-34a-5p in the NAc (**Figure 7D**).

Reviewer #4 (Remarks to the Author):

The authors have included a very detailed response to Reviewers 1-4 in the revised manuscript-which includes some additional data. The responses started off well. While the inclusion of a few

words suggesting that there a direct link between mTorc and miR-34a is requires further investigation in the discussion is appreciated, if this reviewer is still not convinced, but this is a matter of interpretation that should be left for the reader. However, the authors failed to address several comments and concerns raised by reviewer 4 - that remain problematic for publication.

1) the findings for figure 7A are not convincing, and at best, may represent a change in tolerance. Further, in their response, the authors replied that the asterisk signifies the effect of the virus and no interaction of day. Please confirm that this was within subject analysis. To this end, the original comment suggested that the minimal effect was in part due to overexpression using lenti viral system that was not specific for neurons much less D1 or D2 neurons. This has not been addressed and weakens any conclusion that suggest as causative role for cell-type specific manner.

Answer: As mentioned in the response to reviewer 3 comment, to overexpress miR-34a-5p specifically in D1 NAc neurons, we utilized the FLEX-Cre approach and designed and generated an AAV2-Flex-miR-34a-5p virus. Overexpression of the virus in the NAc of D1-Cre mice increased alcohol but not sucrose consumption (**Figure 7B,D**). As the AAV2 serotype permits a neuron-specific expression (**Supplementary Figure 11**), these results suggest a cell-specific action of miR-34a-5p on alcohol intake (**See also Rebuttal Figure 1**).

2) Given the limited and not robust data in figure 7, the question became are the additional data the authors have to demonstrate the role of miR-34a-5p in additional behaviors associated with alcohol use and abuse. Instead of including any measure that may have strengthened these arguments here, the authors included that they have recently obtained a grant. This does not help support the data in this manuscript for Nature Communications.

Answer: We believe that the additional experiments establish the role of miR-34a-5p in D1 neurons in the escalation of excessive alcohol drinking (**Figure 7B**). In addition, we report an alcohol-specific effect; as the consumption of a naturally rewarding substance, sucrose, was unaffected (**Figure 7D**). We could not use alcohol self-administration and related behavior (seeking, extinction, reinstatement) since it requires mice to undergo 20% alcohol 2BC prior to it, which would lead to an endogenous alcohol-dependent increase of miR-34a-5p levels.

Reviewer #4 (Remarks to the Author):

The statistics question and presentation of statistics are not fixed or understood.

1. *How can F values be a decimal?*

Answer: We are puzzled by this comment. F values are decimal. Below is a list of random Science, Nature, Nature Communications and Neuron papers that reported the F values with a decimal¹⁻⁴. We utilize PRISM GraphPad for statistical analysis and report the values generated by the program. However, if the reviewer refers to the Df being decimal, it is due to the Greisser Greenhouse correction performed by Prism. We reanalyzed the data without this correction.

2. *What does mixed measures mean relative to repeated measures?*

Answer: The data set in 7B contains a missing value due to a leaking alcohol bottle during a 24h drinking session. Prism GraphPad uses a mixed-effects model instead of RM Two-way ANOVA when missing values are present.

3. *How can the df of the f values be nearly identical in figure 7b and c given the number of measures and animals are more than 2x as many in figure 7c?*

Answer: We apologize for the oversight; the n shown for Figure 7B is incorrect. The size of the group in Figure 7B is n=9-10.

4. *Why are there no error bars for figure 7c.*

Answer: The standard deviation is shown using lines instead of bars. Figures 7C and 8M have been changed and now show SEM using error bars.

5. *What was the scientific rationale for using a Sidak's post hoc for some graphs and a Tukey's for other figures?*

Answer: We performed the post hoc test recommended by PRISM Graphpad. Specifically, when comparing every row (or column) mean with every other row (or column) mean (performing all possible pairwise comparisons), Tukey's test is performed. When only certain levels need comparison (focusing on one significant main effect), Sidak's test is performed. We added these details in the method section.

6. *There is a general lack of completeness and carefulness in many of the reporting and presentation. While the information is interesting in this manuscript, and much work has gone into this, neither aspects are demonstrated by the way this manuscript or rebuttal(s) are constructed.*

Answer: We do not know how to address this very generalized comment, which we wholeheartedly disagree with and find hurtful.

List of publications reporting F values:

- 1 Augier, E. *et al.* A molecular mechanism for choosing alcohol over an alternative reward. *Science* **360**, 1321-1326, doi:10.1126/science.aao1157 (2018).
- 2 Piatkevich, K. D. *et al.* Population imaging of neural activity in awake behaving mice. *Nature* **574**, 413-417, doi:10.1038/s41586-019-1641-1 (2019).

- 3 Iwata, T. *et al.* Hippocampal sharp-wave ripples correlate with periods of naturally occurring self-generated thoughts in humans. *Nat Commun* **15**, 4078, doi:10.1038/s41467-024-48367-1 (2024).
- 4 Chandra, R. *et al.* Drp1 Mitochondrial Fission in D1 Neurons Mediates Behavioral and Cellular Plasticity during Early Cocaine Abstinence. *Neuron* **96**, 1327-1341 e1326, doi:10.1016/j.neuron.2017.11.037 (2017).

Reviewer #4 (Remarks to the Author):

The statistics question and presentation of statistics are not fixed or understood.

1. *How can F values be a decimal?*

Answer: We are puzzled by this comment. F values contain decimal. Below is a list of random Science, Nature, Nature Communications and Neuron papers that reported the F values with a decimal¹⁻⁴. We utilize PRISM GraphPad for statistical analysis and report the values generated by the program. However, if the reviewer refers to the Df being decimal, it is due to the Greisser Greenhouse correction performed by Prism. We reanalyzed the data without this correction.

2. *What does mixed measures mean relative to repeated measures?*

Answer: The data set in 7b contains a missing value due to a leaking alcohol bottle during a 24h drinking session. Prism GraphPad uses a mixed-effects model instead of RM Two-way ANOVA when missing values are present.

3. *How can the df of the f values be nearly identical in figure 7b and c given the number or measures and animals are more than 2x as many in figure 7c?*

Answer: We apologize for the oversight; the n shown for Figure 7B is incorrect. The size of the group in Figure 7B is n=9-10.

4. *Why are there no error bars for figure 7c.*

Answer: The standard deviation is shown using lines instead of bars. Figures 7C and 8M have been changed and now show SEM using error bars.

5. *What was the scientific rational for using a Sidak's post hoc for some graphs and a Tukey's for other figures?*

Answer: We performed the post hoc test recommended by PRISM Graphpad. Specifically, when comparing every row (or column) mean with every other row (or column) mean (performing all possible pairwise comparisons), Tukey's test is performed. When only certain levels need comparison (focusing on one significant main effect), Sidak's test is performed. We added these details in the method section.

6. *There is a general lack of completeness and carefulness in many of the reporting and presentation. While the information is interesting in this manuscript, and much work has gone into this, neither aspects are demonstrated by the way this manuscript or rebuttal(s) are constructed.*

Answer: We do not know how to address this very generalized comment.

List of publications reporting F values with decimals:

- 1 Augier, E. *et al.* A molecular mechanism for choosing alcohol over an alternative reward. *Science* **360**, 1321-1326, doi:10.1126/science.aao1157 (2018).
- 2 Piatkevich, K. D. *et al.* Population imaging of neural activity in awake behaving mice. *Nature* **574**, 413-417, doi:10.1038/s41586-019-1641-1 (2019).

- 3 Iwata, T. *et al.* Hippocampal sharp-wave ripples correlate with periods of naturally occurring self-generated thoughts in humans. *Nat Commun* **15**, 4078, doi:10.1038/s41467-024-48367-1 (2024).
- 4 Chandra, R. *et al.* Drp1 Mitochondrial Fission in D1 Neurons Mediates Behavioral and Cellular Plasticity during Early Cocaine Abstinence. *Neuron* **96**, 1327-1341 e1326, doi:10.1016/j.neuron.2017.11.037 (2017).